# The olfactory critical period is determined by activity-dependent Sema7A/PlxnC1 signaling within glomeruli

Nobuko Inoue[1], Hirofumi Nishizumi[1], Rumi Ooyama[2], Kazutaka Mogi[2], Katsuhiko Nishimori[3†], Takefumi Kikusui[2], Hitoshi Sakano[1]*

[1]Department of Brain Function, School of Medical Sciences, University of Fukui, Matsuoka, Japan; [2]Department of Animal Science and Biotechnology, School of Veterinary Medicine, Azabu University, Sagamihara, Japan; [3]Department of Molecular and Cell Biology, Graduate School of Agricultural Science, Tohoku University, Sendai, Japan

*For correspondence:
sakano.hts@gmail.com

Present address: †Department of Obesity and Internal Inflammation, Fukushima Medical University, Fukushima, Japan

Competing interests: The authors declare that no competing interests exist.

**Abstract** In mice, early exposure to environmental odors affects social behaviors later in life. A signaling molecule, Semaphorin 7A (Sema7A), is induced in the odor-responding olfactory sensory neurons. Plexin C1 (PlxnC1), a receptor for Sema7A, is expressed in mitral/tufted cells, whose dendrite-localization is restricted to the first week after birth. Sema7A/PlxnC1 signaling promotes post-synaptic events and dendrite selection in mitral/tufted cells, resulting in glomerular enlargement that causes an increase in sensitivity to the experienced odor. Neonatal odor experience also induces positive responses to the imprinted odor. Knockout and rescue experiments indicate that oxytocin in neonates is responsible for imposing positive quality on imprinted memory. In the oxytocin knockout mice, the sensitivity to the imprinted odor increases, but positive responses cannot be promoted, indicating that Sema7A/PlxnC1 signaling and oxytocin separately function. These results give new insights into our understanding of olfactory imprinting during the neonatal critical period.

## Introduction

Mammalian sensory systems are generated by a combination of activity-dependent and -independent processes. The basic architecture of sensory systems is built before birth based on a genetic program (*Hong and Luo, 2014*; *Luo et al., 2008*; *Yogev and Shen, 2014*). However, the sensory maps and neural circuits are further refined in an activity-dependent manner (*Espinosa and Stryker, 2012*; *Hensch, 2005*; *Hooks and Chen, 2007*; *Kirkby et al., 2013*; *Okawa et al., 2014*). In the mouse olfactory system, a coarse map is generated independently from neuronal activity (*Mori and Sakano, 2011*). Axon targeting along the dorsal-ventral (D-V) axis is regulated by positional information of olfactory sensory neurons (OSNs) (*Takeuchi et al., 2010*). In contrast, targeting along the anterior-posterior (A-P) axis is instructed by odorant receptor (OR) molecules using cAMP as a second messenger (*Imai et al., 2006*). Each OR species generates a specific level of cAMP by intrinsic, non-neuronal OR-activity (*Nakashima et al., 2013*) that determines expression levels of A-P targeting molecules including Semaphorin (Sema) 3A, Neuropilin (Nrp) 1, and Plexin (Plxn) A1 (*Imai et al., 2009*). The map is then refined by axon-sorting molecules, for example, Kirrel 2/3 and Ephrin A/eph A receptors (*Serizawa et al., 2006*) whose expression is regulated by neuronal activity using cAMP generated by olfactory G protein ($G_{olf}$) and cyclic-nucleotide-gated (CNG) channels (*Sakano, 2010*). It has been reported that this activity for the map refinement is intrinsic and not odor-evoked (*Nakashima et al., 2019*). Intrinsic OSN activity is also needed for synapse formation within

glomeruli. *Inoue et al., 2018* found that odor-evoked OSN activity further promotes synapse formation and dendrite selection in mitral/tufted (M/T) cells using Sema7A/PlxnC1 signaling.

In neonates, there is a narrow time frame, referred to as the critical period, that allows proper development in response to environmental inputs. This activity-dependent process is initially plastic but soon becomes irreversible. If the circuit is left unstimulated, the brain function served by that circuit is impaired. Thus, sensory inputs during the critical period are important to make the system functional. Unlike other sensory systems, the olfactory system constantly regenerates OSNs throughout the animal's life span (*Costanzo, 1991*; *Graziadei and Monti Graziadei, 1985*; *Leung et al., 2007*). Although OSNs are replaced and form new connections with second-order neurons, proper circuits cannot be regenerated once the existing OSNs are completely ablated after the early neonatal period. This time frame of olfactory circuit formation has been referred to as the olfactory critical period (*Ma et al., 2014*; *Tsai and Barnea, 2014*). However, the olfactory critical period has not been defined at the molecular level, and the key process that makes this time-window critical has yet to be clarified.

In mammals, neonatal exposure to environmental odorants can affect odor perception and behavior later in life (*Logan et al., 2012*; *Mennella et al., 2001*; *Sullivan et al., 2000*; *Wilson and Sullivan, 1994*). Salmon and trout return to their home river based on olfactory memory (*Nevitt et al., 1994*). In the visual system, ducklings follow the first moving object upon hatching recognizing it as the parent bird (*Horn, 1998*; *Nakamori et al., 2013*). Although such phenomena are widely known, little is known about how imprinting is established and how the imprinted memory modulates innate behavioral decisions.

In an effort to address these unanswered questions about the olfactory critical period, we studied olfactory imprinting in mice by analyzing a pair of signaling molecules, Sema7A and PlxnC1, which induces post-synaptic events within glomeruli. We found that Sema7A/PlxnC1 signaling enlarges glomeruli resulting in increased sensitivity to experienced odors. Neonatal odor experience also induces positive responses to the imprinted odor memory. Separately from Sema7A/PlxnC1 signaling, oxytocin in neonates is responsible for imposing the positive quality on imprinted memory.

## Results

### Naris occlusion/reopen experiments

In order to precisely determine the critical period that causes plastic changes in synapse formation and odor detection, we performed naris occlusion/reopen experiments (*Figure 1A* and *Figure 1—figure supplement 1*). Unilateral naris occlusion at postnatal day 0 (P0) was continued through various time points, and resultant synapse markers within glomeruli were examined at P21. We found that the levels of a pre-synapse marker vGlut2 (*Honma et al., 2004*) and a post-synapse marker GluR1 (*Montague and Greer, 1999*), were normal when the occluded naris was reopened at P6 or before. However, both markers decreased and remained low when the occluded naris was reopened after P8.

We then studied the effect of naris occlusion on odor detection in the habituation/dishabituation test (*Figure 1B*). The ability to sense odors using the occluded naris was tested by plugging the non-occluded naris at the 6th week of life (6w). When the occluded naris was reopened at P6, mice demonstrated normal abilities to detect odor information. In contrast, when the occlusion was continued to P10, the mice demonstrated reduced responsiveness to odorants and poor ability to discriminate enantiomers, (+) and (–) carvones (CARs). Taken together, the critical period in the mouse olfactory system appears to be restricted to the early neonatal period, approximately during the first week after birth.

### Odor perception is specifically affected by the odorants exposed as neonates

In order to study whether neonatal odor experience can affect odor perception, we exposed pups to a particular odorant and analyzed their responses as adults. In the habituation/dishabituation test at 6w, the mice conditioned to vanillin (VNL) at P2~4 demonstrated increased responsiveness/sensitivity to VNL (*Figure 2*). Furthermore, unlike the unconditioned control, the VNL-conditioned mice showed lasting interests for VNL and were not fully habituated when VNL was presented a second

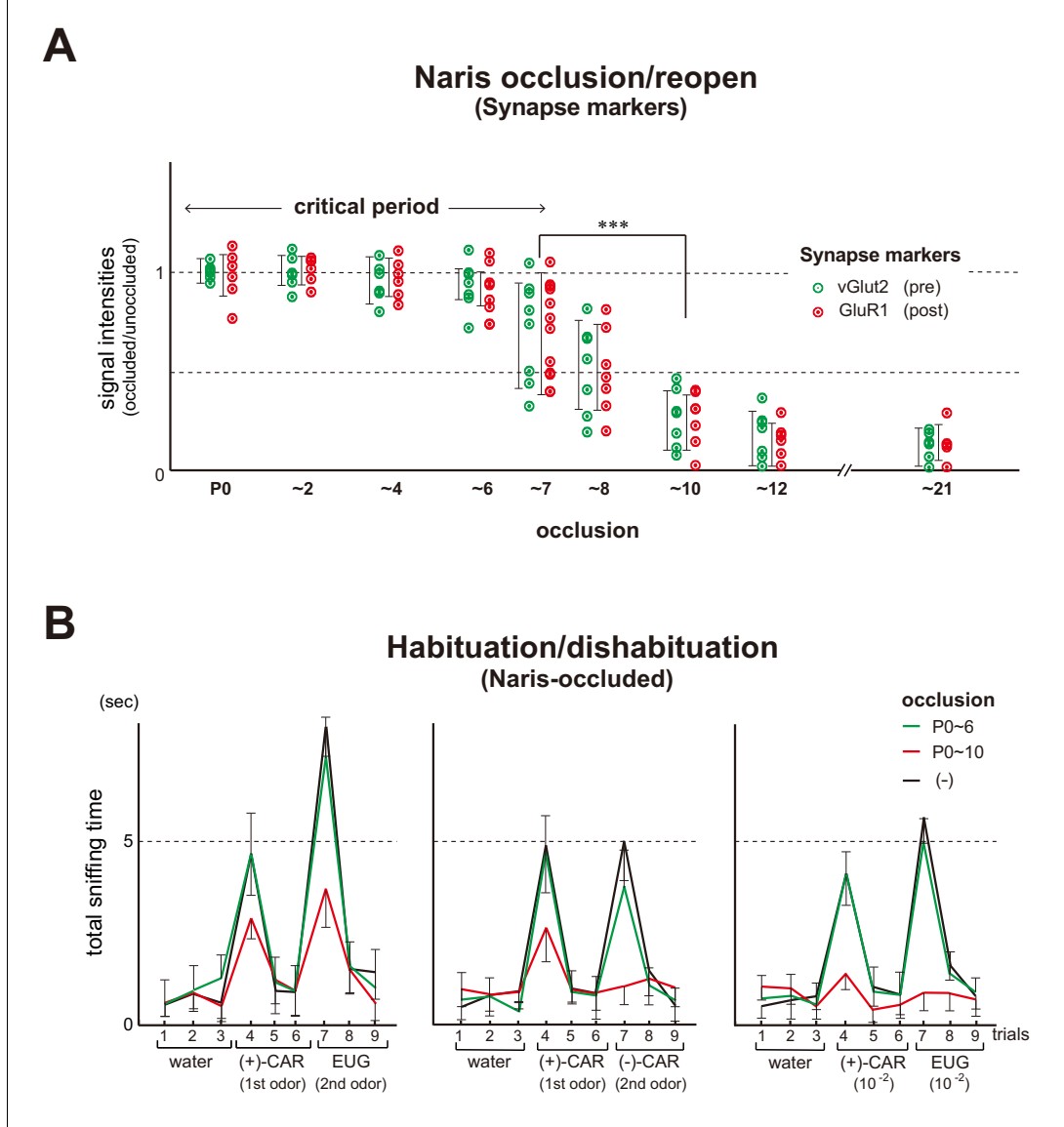

**Figure 1.** Unilateral naris occlusion in the mouse neonates. (**A**) Effects of naris occlusion on the expression of synapse markers within glomeruli. Mice were unilaterally occluded at P0 and the occluded naris was reopened at various time points. OB samples were isolated at P21 and analyzed by immunostaining for pre- and post-synapse markers, vGlut2 and GluR1, respectively. Relative staining levels of these markers in the glomerular layer are compared. Error bars are SD (n = 10 glomeruli for each condition). See also *Figure 1—figure supplement 1*. (**B**) Odor detection in the naris-occluded mice. Six-week-old mice used in this assay were unilaterally naris-occluded at P0 and the occluded naris was reopened at P6 (green) or P10 (red). Mice without occlusion (–) were analyzed as controls (black). Mice were habituated to the cage and then a filter paper spotted with 0.5 µl of distilled water was presented for 3 min. This was repeated three times with 1 min intervals (control trials 1–3). Next, a filter paper spotted with the 1st odor was presented three times (detection trials 4–6). Then, a filter paper spotted with the 2nd odor was presented three times (detection trials 7–9). Investigation times (sec) observed during each presentation were measured. Odorant pairs examined were as follows: left, 6.2 M (+)-CAR and 6.4 M EUG; middle, 6.2 M (+)-CAR and 6.3 M (–)-CAR; and right, 62 mM (+)-CAR and 64 mM EUG. Error bars are SD (n = 15, 12, 10 animals, 10 litters). ***p<0.005 (Student's *t*-test). n.s., not significant. CAR, carvone; EUG, eugenol.

The online version of this article includes the following source data and figure supplement(s) for figure 1:

**Source data 1.** Odor detection in the naris-occluded mice.
**Figure supplement 1.** Effects of naris occlusion on the expression of synapse markers within glomeruli.

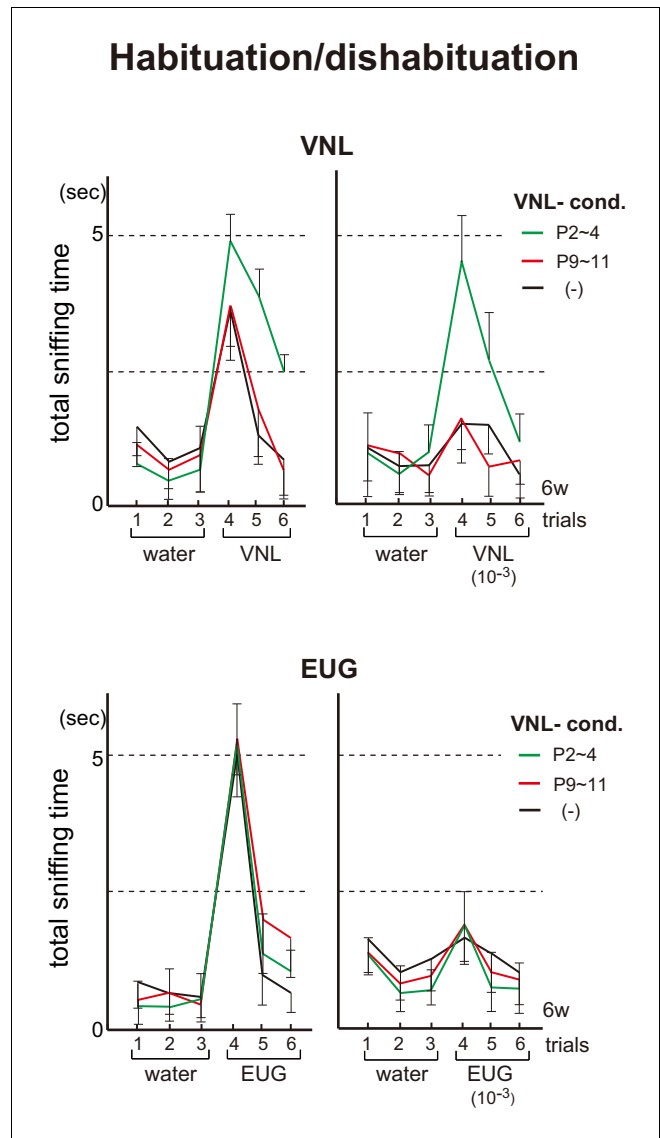

**Figure 2.** Effects of neonatal odor experience on odor perception in adults. Odor detection in the vanillin (VNL)-conditioned mice. Mice were habituated to the cage and then a filter paper spotted with 0.5 μl of distilled water was presented for 3 min. This was repeated three times with 1 min intervals (control trials 1–3). Then, a filter paper spotted with 0.5 μl of 20 mM VNL or 6.4 M eugenol (EUG) was presented three times (detection trials 4–6). Investigation times (sec) observed during each presentation were measured. Mice were conditioned to VNL at P2~4 (green) or P9~11 (red), and were analyzed as adults at 6w. Mice without VNL conditioning (–) were analyzed as controls (black). Odorants with $10^{-3}$ dilution were also analyzed and shown in the right of each figure set. Error bars are SD (n = 5, 6, 7 animals, 5 litters).

The online version of this article includes the following source data for figure 2:

**Source data 1.** Odor detection in the vanillin (VNL)-conditioned mice.

time. Such conditioning effects were neither found in mice conditioned to VNL at P9~11, nor observed toward other unconditioned odorants, for example, eugenol (EUG).

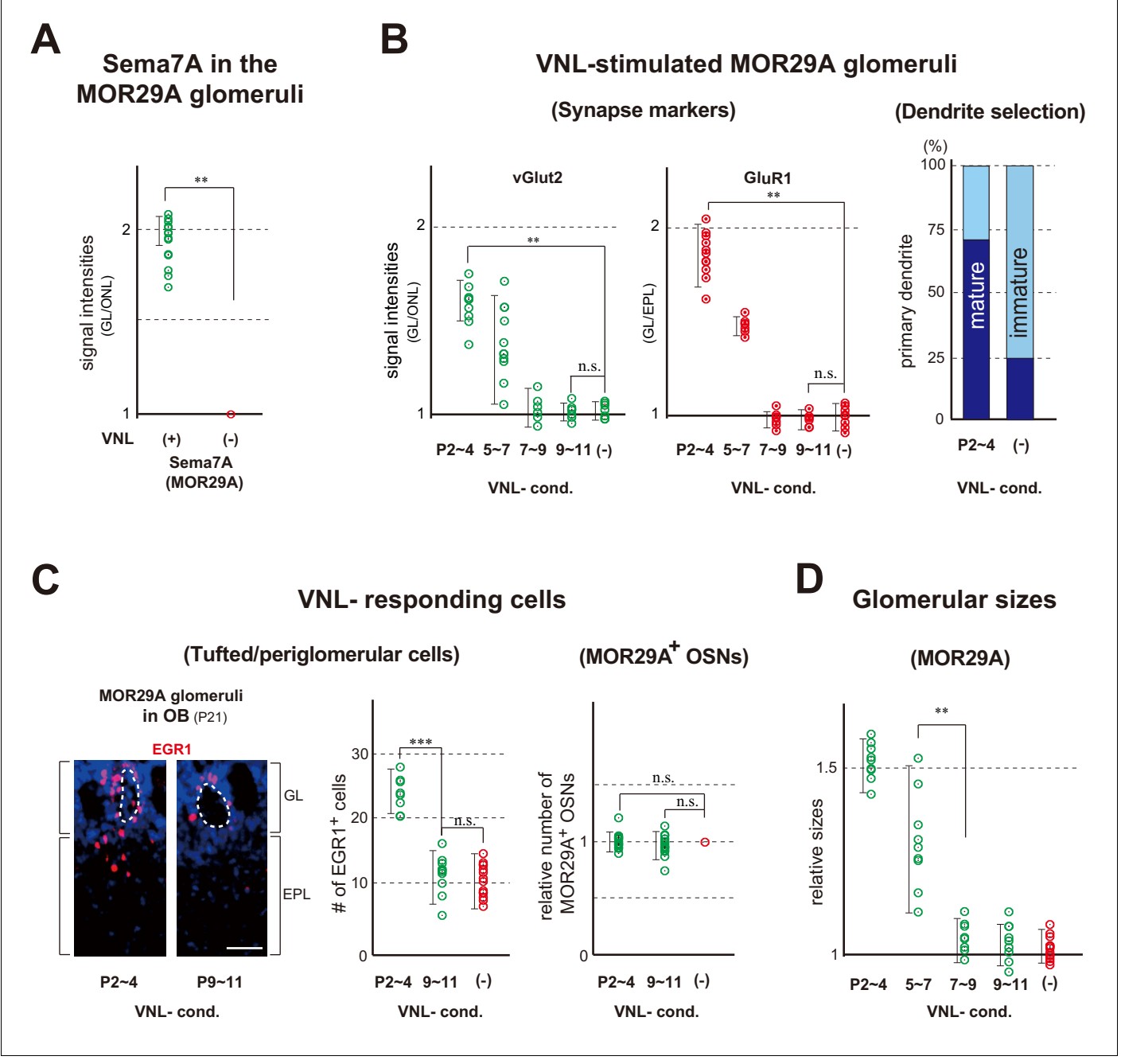

**Figure 3.** Glomerular changes by odor exposure in neonates. (**A**) Sema7A expression stimulated by VNL exposure. The ECFP-tagged MOR29A glomeruli were analyzed by immunohistochemistry after the exposure to VNL at P2~4. OB sections were immunostained at P4 using antibodies against Sema7A. Relative fluorescent signals within the MOR29A glomeruli were analyzed in the mice with (+) or without (–) VNL exposure. **p<0.01 (Student's t-test). Error bars are standard deviation, SD (n = 6 animals for each condition). See also *Figure 3—figure supplement 1A*. (**B**) MOR29A glomeruli in the VNL-exposed mice. Left: Changes in the levels of synapse markers. Pups were exposed to VNL at different time periods, and the OB sections at P11 were immunostained for pre- and post-synapse markers, vGlut2 and GluR1, respectively. Signal intensities within the glomerular layer (GL) were normalized by those detected in the olfactory nerve layer (ONL) or external plexiform layer (EPL), and compared with the VNL-unexposed controls (–). **p<0.01 (Student's t-test). n.s., not significant. Error bars indicate SD (n = 6, 3, 3, 5, 5 glomeruli for each condition). See also *Figure 3—figure supplement 1B*. Right: Dendrite selection within the MOR29A glomeruli. The mice conditioned to VNL (P2~4) and unconditioned (–) were analyzed. M/T cells at P4 were visualized by Lucifer yellow (LY) injection into the glomeruli (*Figure 3—figure supplement 2*). Intracellular LY injection was performed as previously described (*Inoue et al., 2018*). The numbers of M/T cells with one dendrite (mature) and those with multiple dendrites (immature) were counted in the MOR29A glomeruli. The ratios (%) of mature (dark blue) and immature (cyan) M/T cells are shown: VNL-cond., 12/17 (70.6 %); VNL-uncond., 4/16 (25.0 %). n = 6, 5 glomeruli. See also *Figure 3—figure supplement 2*. (**C**) VNL-responding cells connecting to the MOR29A glomeruli.
*Figure 3 continued on next page*

*Figure 3 continued*

Left: Detection of tufted and periglomerular cells activated by VNL. The MOR29A glomeruli were identified by ECFP signals. The sections were then counter-stained with DAPI. Levels of an immediate-early gene product EGR1 (red) surrounding the MOR29A glomeruli were analyzed at 3w by immunostaining of OB sections (25 μm-thick). The mice were conditioned to VNL at P2~4 or P9~11. Middle: Quantification of EGR1 signals. EGR1 signals in the glomerular layer (GL) and external plexiform layer (EPL) were counted as activated periglomerular cells and tufted cells, respectively. Mice without VNL exposure in neonates were analyzed as controls (–). Error bars are SD (n = 4, 4, 5 animals). The one-way ANOVA was applied on values. ***$p<0.005$ (Student's *t*-test). n.s., not significant. Scale bar is 30 μm. Right: MOR29A-positive OSNs. ECFP-tagged MOR29A OSNs in the OE were counted at P21 after the exposure to VNL at P2~4 or P9~11. To identify the MOR29A-positive OSNs, OE sections were immunostained with antibodies against GFP. Relative numbers of MOR29A[+] OSNs are compared for the mice with (P2~4, 9~11) or without (–) VNL exposure. n.s., not significant. Error bars are standard deviation, SD (n = 6, 5, 4 animals). See also *Figure 3—figure supplement 1C*. (D) Glomerular sizes with or without VNL conditioning. Relative sizes (ratios of diametral areas) of the MOR29A glomeruli were measured at 3w after the VNL exposure at different time periods in neonates. Glomeruli without VNL exposure (–) were analyzed as negative controls. Error bars are SD (n = 6, 3, 3, 5, 5 animals). VNL, vanillin; GL, glomerular layer; ONL, olfactory nerve layer; EPL, external plexiform layer.

The online version of this article includes the following source data and figure supplement(s) for figure 3:

**Figure supplement 1.** MOR29A glomeruli in the VNL-exposed mice.
**Figure supplement 2.** Lucifer yellow injection.
**Figure supplement 2—source data 1.** Dendrite selection within the MOR29A glomeruli.
**Figure supplement 3.** Changes within the MOR29A glomeruli in the naris-occluded mice.

## Odor exposure promotes dendrite selection resulting in glomerular enlargement

As neonatal odor experience affects odor perception in adults, we examined whether any changes could be found in the glomeruli in response to odor exposure. We focused on an OR, MOR29A (also known as Olfr1510), whose ligand is known as VNL (*Tsuboi et al., 2011*). This is because the onset of its expression is relatively early during olfactory development and its intrinsic level of Sema7A is low compared with other OR species. Exposure to VNL at P2~4 increased Sema7A expression (*Figure 3A* and *Figure 3—figure supplement 1A*), which promotes post-synaptic events and dendrite selection in mitral/tufted (M/T) cells (*Inoue et al., 2018*). VNL stimulation also increased the levels of synaptic markers, vGlut2 and GluR1 (*Figure 3B* left and *Figure 3—figure supplement 1B*). We then examined dendrite maturation by injecting Lucifer yellow into the glomeruli to visualize a single connecting M/T cell (*Figure 3—figure supplement 2*). The selection of primary dendrites was promoted within the MOR29A glomeruli (*Figure 3B* right). In this experiment, M/T cells with a selected primary-dendrite were classified as mature, whereas those with multiple unselected dendrites in the glomerular layer were classified as immature.

We found that sizes of the MOR29A glomeruli increased with a ligand odor as previously reported for another OR, M72 (also known as Olfr160) (*Liu et al., 2016*; *Todrank et al., 2011*). When the pups were exposed to VNL at P2~4, the numbers of VNL-responsive cells surrounding the MOR29A glomeruli increased at P21 (*Figure 3C* left and *Figure 3—figure supplement 1C*), resulting in the enlargement of MOR29A glomeruli (*Figure 3D*). Interestingly, the number of MOR29A-expressing OSNs remained the same (*Figure 3C* right). These changes were not found when VNL was exposed at P7~9 or P9~11.

It is notable that in the olfactory bulb (OB) of naris-occluded mice, the levels of synaptic markers in the MOR29A glomeruli were lowered (*Figure 3—figure supplement 3* left) and the glomerular sizes for MOR29A was reduced (*Figure 3—figure supplement 3* middle). Furthermore, VNL-responding tufted cells and periglomerular cells connecting to the MOR29A glomeruli were reduced (*Figure 3—figure supplement 3* right). These changes were observed when occlusion was performed at P0~4, but not at P7~11 (*Figure 3—figure supplement 3* right). The time frame that allows for activity-dependent changes appears to be restricted to the early neonatal period, roughly during the first week after birth. It should be noted that the critical period as determined by naris occlusion, represents the maximum time frame covering for all OR species. In contrast, in the odor exposure experiment, the critical period is determined for a specific OR species and can be different among various ORs depending upon the onset of their expression.

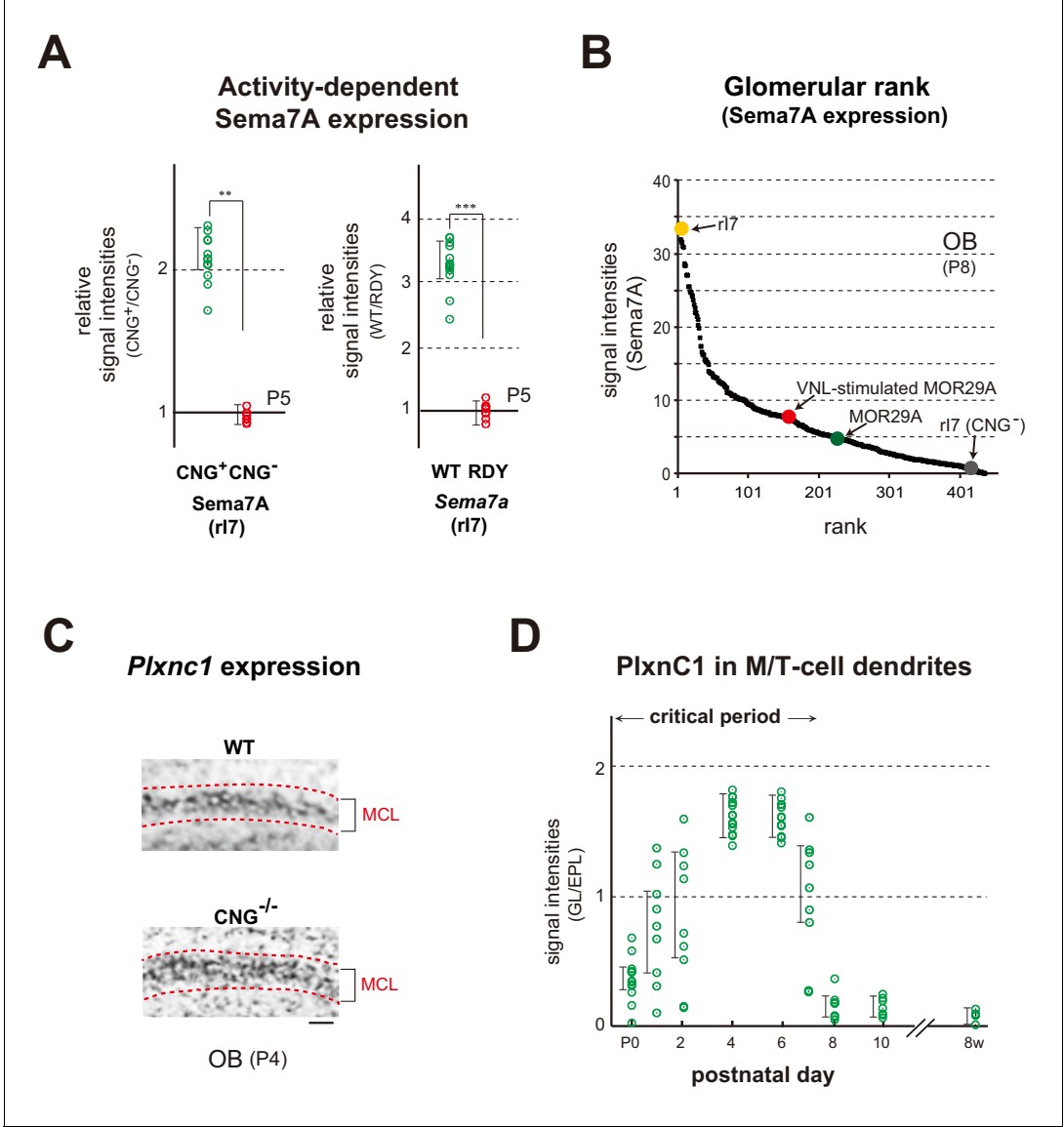

**Figure 4.** Expression of Sema7A and PlxnC1 in the neonatal OB. (**A**) Activity-dependent expression of Sema7A. Left: Analysis of the CNG-A2$^{+/-}$ mice. Duplicated glomeruli of rI7 were analyzed for Sema7A expression in the CNG-A2$^{+/-}$ female mice at P5. EYFP-tagged rI7 glomeruli were detected by immunostaining with anti-GFP antibodies. OB sections were immunostained with antibodies against CNG-A2 and Sema7A. Relative signal intensities of Sema7A (CNG$^+$/CNG$^-$) are compared between the CNG$^+$ and CNG$^-$ glomeruli. **p<0.05 (Student's $t$-test). Error bars indicate SD (n = 8 animals). Right: Effects of the DRY-motif mutation of OR on Sema7A expression. OE sections expressing the WT and DRY-motif mutant (RDY) of rI7 were analyzed by in situ hybridization for *Sema7a* transcripts at P5. The DRY-motif mutant suppresses *Sema7a* transcription because the mutant receptor does not produce cAMP that generates CNG-channel activity. OSNs expressing the EYFP-tagged rI7 were detected by immunostaining with anti-GFP antibodies. Relative signal intensities of *Sema7a* transcripts (WT/DRY) are compared between the WT and DRY-motif mutant of rI7. ***p<0.005 (Student's $t$-test). Error bars indicate SD (n = 4 animals). See also *Figure 4—figure supplement 1A*. (**B**) Ranking of glomeruli for Sema7A expression. Individual glomeruli possess unique but different levels of Sema7A expression determined by intrinsic activity of ORs, forming the glomerular rank of Sema7A expression. OB sections were immunostained with anti-Sema7A antibodies. Intensities of Sema7A signals were determined for each glomerulus and plotted in order. Glomerular rank of fluorescent signals is shown for 437 different glomeruli in the OB at P8. Expression levels of Sema7A are indicated for the rI7 (yellow), MOR29A (green), VNL-stimulated (P5~7) MOR29A (red), and CNG$^-$ rI7 glomeruli (gray). See also *Figure 3—figure supplement 1A* and *Figure 4—figure supplement 1A*. (**C**) Expression of *Plxnc1* in the OB. Both CNG$^{+/+}$ and CNG$^{-/-}$ mice were analyzed. OB sections were analyzed at P4 by in situ hybridization using the *Plxnc1* probe. Mitral cell layers (MCL) are circled by dotted lines. n = 6 animals. Scale bar, 20 µm. See also *Figure 4—figure supplement 1B*. (**D**) Localization of PlxnC1 in the M/T-cell dendrites. To detect PlxnC1, a receptor for Sema7A, OB sections were immunostained with PlxnC1 antibodies. Relative signal intensities (GL/EPL) are shown for different time points in the neonatal period. Note that PlxnC1 is found in the M/T-cell dendrites only during the first week after birth. n = 2 animals except for P4 (n = 6). GL, glomerular layer; EPL, external plexiform layer.

*Figure 4 continued on next page*

*Figure 4 continued*

The online version of this article includes the following source data and figure supplement(s) for figure 4:

**Source data 1.** Ranking of glomeruli for Sema7A expression.
**Figure supplement 1.** Activity-dependent expression of Sema7A in OSNs and temporal localization of PlxnC1 in M/T-cell dendrites.

## Sema7A/PlxnC1 signaling defines the olfactory critical period

We next proceeded to study the key process that causes the neonatal time frame to become critical. Our previous study (*Inoue et al., 2018*) indicated that post-synaptic events and dendrite selection in M/T cells are regulated by a pair of signaling molecules, Sema7A (*Pasterkamp et al., 2003*) and its receptor PlxnC1 (*Tamagnone et al., 1999*) expressed in OSN axons and M/T-cell dendrites, respectively. We analyzed the cell-type-specific conditional knockout (cKO) of PlxnC1 for synapse formation. The M/T-cell-specific cKO was generated using the *Pcdh21* promoter for the Cre driver. We also analyzed the Sema7A total knockout (KO) in parallel. It was found that post-synaptic density (PSD) is rarely formed at P5, however, targeting of OSN axons is not affected in these KOs (*Inoue et al., 2018*).

Expression of the *Sema7a* gene appears to be regulated in an activity-dependent manner, as Sema7A is diminished in the KO of CNG channels (*Brunet et al., 1996*) that generate OR-specific OSN activity (*Figure 4A* left and *Figure 4—figure supplement 1A* left). Supporting this observation, the mutant OR defective in G-protein coupling (*Imai et al., 2006*), also diminished *Sema7a* expression (*Figure 4A* right and *Figure 4—figure supplement 1A* right). Individual glomeruli possess unique but different levels of Sema7A expression, determined by the intrinsic activity of ORs, thereby forming the glomerular rank of Sema7A. Sema7A expression increased with a ligand odor and the glomerular rank of MOR29A became higher for Sema7A by VNL exposure (*Figure 4B*). In contrast to Sema7A, PlxnCl expression or its localization to the M/T-cell dendrites was not affected in the KO of Sema7A or CNG-A2 (*Figure 4C* and *Figure 4—figure supplement 1B*). Interestingly, however, its localization to the tuft structure of M/T cells was restricted to the first week after birth (*Inoue et al., 2018*; *Figure 4D*). Thus, Sema7A expression in OSNs supports the activity dependency, and PlxnC1 localization in M/T-cell dendrites limits the time frame of plastic changes within glomeruli during the olfactory critical period.

## Sema7A is sufficient to induce post-synaptic events in M/T cells

We wanted to determine whether there are any additional components for synapse formation, whose expression is downstream of OR-specific OSN activity. To address this question, we performed rescue experiments in the CNG-A2 KO. In the KO mice, synapse formation is significantly affected (*Figure 5—figure supplement 1*). Since the *Cnga2* gene is located on chromosome X, duplicated glomeruli are formed in the hemizygous KO (*Zhao and Reed, 2001*) due to stochastic X-chromosome inactivation. In the CNG-negative glomeruli of rI7 (also known as *Lofr226*), both pre- and post-synaptic markers decreased and primary-dendrite selection was delayed in contrast to the CNG-positive glomeruli (*Figure 5A*). We then introduced the transgenic (Tg) *Sema7a* gene with the activity-independent promotor of MOR23 (also known as *Olfr16*) into the *Cnga2*$^{+/-}$ female mice with the Sema7A KO background. When the rI7 glomeruli were analyzed, constitutive expression of the Tg *Sema7a* alone restored synapse formation and dendrite selection in the CNG-negative glomeruli (*Figure 5B*). Compared with the endogenous Sema7A in the WT mice, expression levels of Tg-Sema7A were approximately 10% within the rI7 glomeruli in the *Cnga2*$^{+/-}$ mice with the Sema7A KO background (*Figure 5—figure supplement 2A*). The mutant Sema7A (Y213S) (*Liu et al., 2010*), which is incapable of interacting with PlxnC1, did not rescue the KO phenotype of the CNG channels (*Figure 5C* and *Figure 5—figure supplement 2B*). We, therefore, conclude that Sema7A-PlxnC1 interaction is sufficient to induce the activity-dependent post-synaptic events in M/T cells.

## Blockage of Sema7A/PlxnC1 signaling affects social responses later in life

We then studied how mice are affected when Sema7A/PlxnC1 signaling is blocked. Since the Sema7A KO is a total KO and would not be appropriate for behavioral analyses (*Carcea et al., 2014*), we chose the M/T-cell-specific cKO of PlxnC1. Absence of *Plxnc1* expression was confirmed

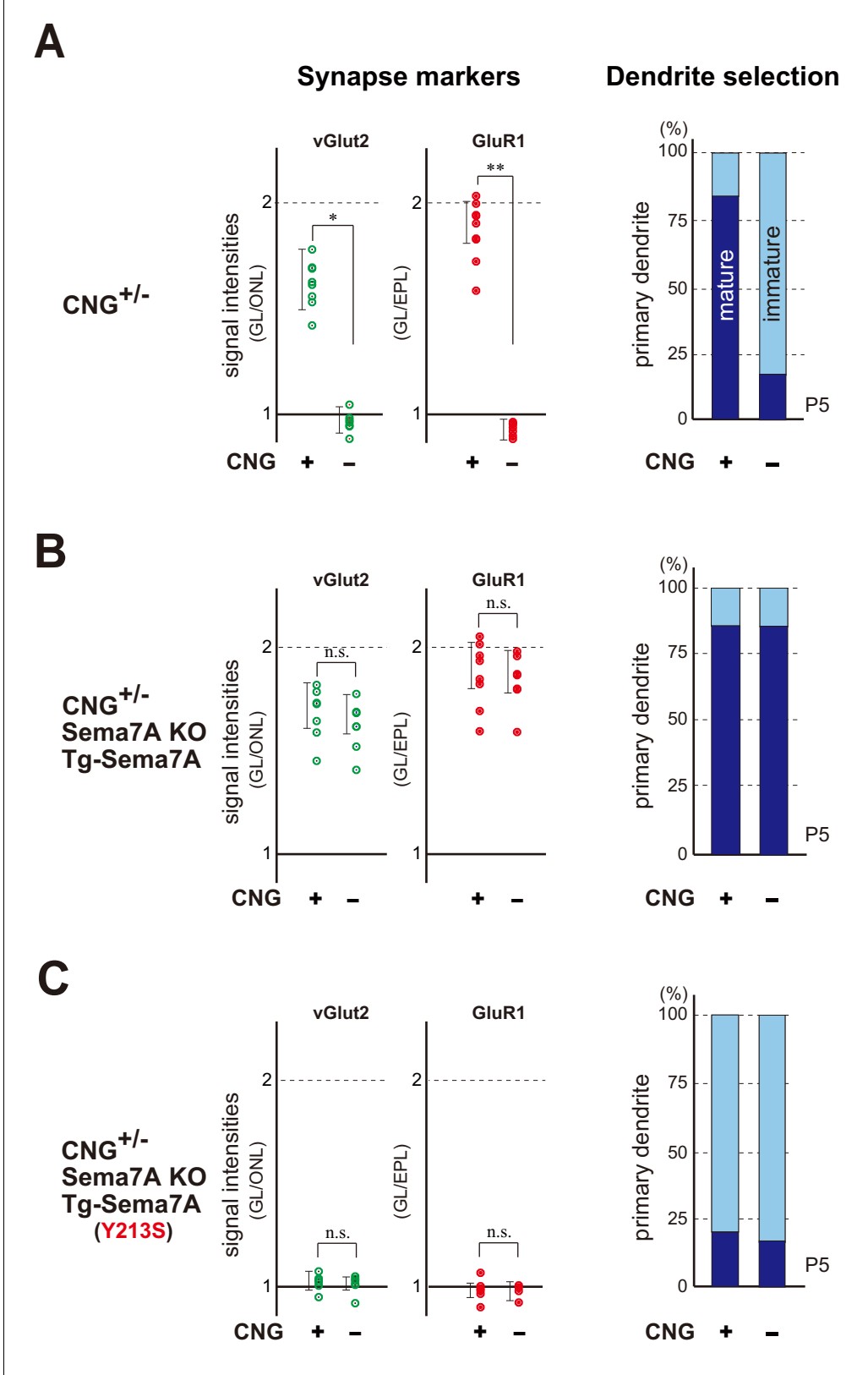

**Figure 5.** Sema7A is a key to synapse formation within glomeruli. (**A**) Synapse markers in the hemizygous female KO of CNG-A2. Duplicated glomeruli (CNG$^+$ and CNG$^-$) of rI7 were analyzed at P5 (n = 8 pairs). Levels of both pre- and post-synapse markers (vGlut2 and GluR1) were reduced in the CNG$^-$ glomerulus compared with the CNG$^+$ (left). Dendrite selection was delayed in the CNG$^-$ glomerulus (right). Dendrites were visualized by LY injection as

*Figure 5 continued on next page*

*Figure 5 continued*

shown in *Figure 3—figure supplement 2*. The ratios of M/T cells with one primary dendrite (mature) and those with multiple branched dendrites (immature) are compared in the rI7 glomeruli: CNG[+], 14/17 (82.3 %); CNG[−], 3/14 (21.4 %). See also *Figure 5—figure supplement 1*. (B) Rescue of synapse formation in the CNG-A2[+/−] mice. The Tg *Sema7a* gene was introduced into the hemizygous female mice of CNG-A2 with the Sema7A KO background. Activity-independent, constitutive expression of the Tg *Sema7a* was sufficient to rescue the defective phenotype of synapse formation (left) and dendrite selection (right) in the CNG-A2 KO glomeruli (n = 6 pairs). The ratios of mature and immature M/T cells are: Tg-Sema7A, CNG[+], 15/18 (83.3 %); Tg-Sema7A, CNG[−], 13/16 (81.3 %). (C) Rescue experiments with the interaction mutant of Sema7A in the CNG- A2[+/−] mice. The mutant Sema7A (Y213S) incapable of binding to PlxnC1 was used for the rescue experiment (n = 5 glomeruli). *p<0.05, **p<0.01 (Student's *t*-test). n.s., not significant. The ratios of mature and immature M/T cells are: Tg-Sema7A (Y213S), CNG[+], 3/13 (23.0 %); Tg-Sema7A (Y213S), CNG[−], 2/9 (22.2 %). n = 8, 7 glomeruli. For the synapse markers, relative signal intensities were calculated for the glomerular layer (GL) in comparison to those in the olfactory nerve layer (ONL) or external plexiform layer (EPL). See also *Figure 5—figure supplement 2*.

The online version of this article includes the following source data and figure supplement(s) for figure 5:

**Source data 1.** Dendrite selection within the rI7 glomeruli.
**Figure supplement 1.** Synapse formation in the CNG-A2[−/−] mice.
**Figure supplement 2.** Activity-dependent synapse formation mediated by Sema7A signaling.

by in situ hybridization and immunostaining in the PlxnC1 cKO (*Inoue et al., 2018*). Similar to naris-occluded mice (*Figure 1B*), odor detection was lowered (*Figure 6A*), but not entirely abolished because the cKO mice remained attracted to food smells (*Figure 6B*). It appears that the basic olfactory circuitry is established by intrinsic OSN activity separately from odor-evoked activity (*Gire et al., 2012*).

In the PlxnC1 cKO, social responses in adults were perturbed. While male mice normally demonstrate strong curiosity for unfamiliar mouse scents of both genders, the cKO showed avoidance behavior toward them (*Figure 6B*). A similar impairment in social interactions was observed in the three-chamber test, a standard method to detect autism spectrum disorders (ASD) in rodents (*Moy et al., 2004*). In the three-chambered box, the mouse being tested was placed in a center chamber, an unfamiliar mouse in a plastic cage was to the left, and an empty cage was to the right (*Figure 6C*). Time duration spent in each room was measured during a 15-min test period. Although the WT mouse spent most of its time in the left chamber inspecting the stranger, the PlxnC1 cKO spent longer times in the right chamber, thereby avoiding social interactions with the unfamiliar mouse. These results indicate that Sema7A/PlxnC1 signaling is involved in establishing positive imprinted memory in neonates, whose defect causes the impairment of social responses in adulthood. While the cKO mice fail to demonstrate positive interest toward the odor of unfamiliar mice, they are still able to detect food smells.

Mice demonstrate stress responses in an unfamiliar environment. When they are transferred to a new cage, the rectal temperature rises and remains elevated for about 40 min (*Spooren et al., 2002*). However, in the mice conditioned to VNL at P2~4, stress was eased by VNL at 6w (*Figure 6D*). We tested propionic acid (PPA) as a positive control, whose quality is innately attractive (*Figure 6D*). PPA lowers the stress, but its effect is more moderate compared with the imprinted VNL whose quality is innately neutral. It is notable that such a conditioning effect was not observed in the mice exposed to VNL after the critical period at P9~11. In the PlxnC1 cKO, stress was not eased by the conditioned odor VNL in the hyperthermia test (*Figure 6D*).

## Oxytocin in neonates is needed for smooth social interactions in adults

Increased sensitivity is a form of imprinted memory that can be established at the level of the glomeruli, and its blockage in the PlxnC1 cKO can be explained by the failure of odor-evoked synapse formation. However, lasting interest (*Figure 2*) or easing of stress (*Figure 6D*) is another example of conditioned memory that needs to be further formed in the central brain; a neuronal level not affected by the PlxnC1 cKO. How is it then that the perceptive change takes place for the exposed odors? What is responsible for establishing smooth social interactions by imposing the positive quality on imprinted odor memory? We therefore, turned our attention to various neuronal hormones, including norepinephrine, dopamine, and oxytocin; all of which are known to induce positive mental status and mediate attractive social behaviors (*Andersson, 2000*; *de Wied et al., 1993*). Among them, oxytocin appeared to be promising to us because it is highly expressed in the neonatal brain (*Sannino et al., 2017*) and promotes attractive social interactions (*Bosch and Young, 2018*;

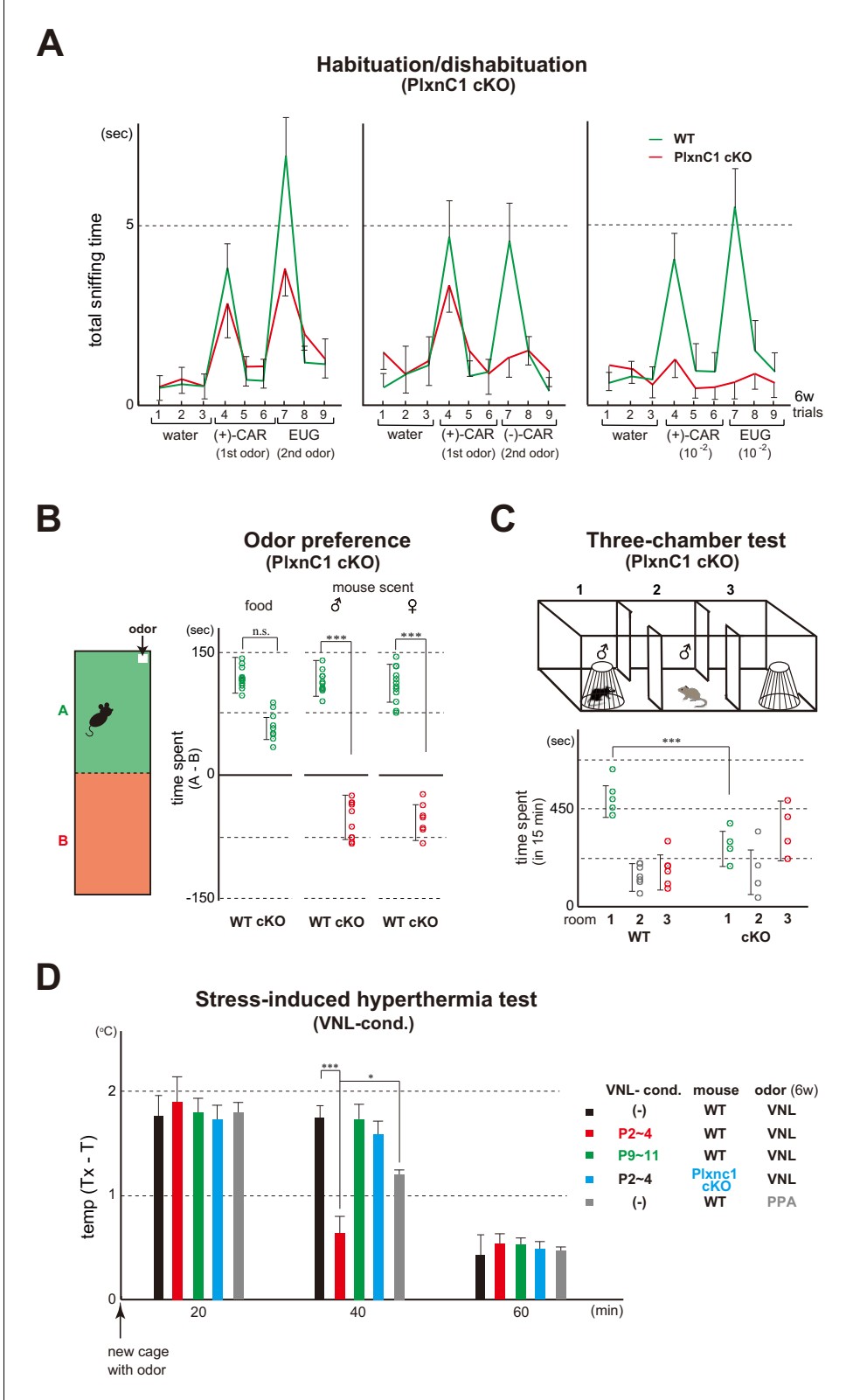

**Figure 6.** Blockage of Sema7A/PlxnC1 signaling affects social behaviors later in life. (**A**) Odor detection of the PlxnC1 cKO in the habituation/dishabituation test. Investigation times for odors were measured in the PlxnC1 cKO and WT male mice at 6w. Please see the legend to *Figure 1B* for the experimental procedure. Odorant pairs examined are as follows: left, 6.2 M (+)-CAR and 6.4 M EUG; middle, 6.2 M (+)-CAR and 6.3 M (–)-CAR; and right, 62 mM (+)-CAR and 64 mM EUG. Error bars are SD (n = 15, 6 animals, 6 litters). CAR, carvone; EUG, eugenol. (**B**) Social responses in the PlxnC1

*Figure 6 continued on next page*

*Figure 6 continued*

cKO. Beds with the odor of unfamiliar mice were presented to the 6w-old WT male or to the PlxnC1 cKO where Sema7A signaling is blocked. The samples were presented in a plastic cup to avoid direct contacts with the mouse nose. Times (sec) spent in the room with or without a test odorant were measured during the 5-min test period. Differences of staying times in the two rooms (A - B) are shown. Mouse food (SLC Japan, Inc) and fresh beds without the mouse scents were used as positive and negative controls, respectively. Error bars are SD (n = 18, 7 animals, 7 litters). (C) Three-chamber test. The male WT and PlxnC1 cKO were analyzed at 6w. The mouse being tested was placed in the center chamber (2). An empty cage was placed to the right (3) and an unfamiliar male mouse in a plastic cage was to the left (1). Time duration (sec) spent in each room was measured during a 15-min test period. Error bars are SD (n = 5, 4 animals, 4 litters). (D) Stress-induced hyperthermia test in the VNL-conditioned PlxnC1 cKO. Pups of the WT (red and green) and PlxnC1 cKO (light blue) were exposed to VNL at P2~4 or P9~11 and analyzed at 6w. Immediately after the transfer to a new cage, a filter paper spotted with VNL was presented to the mice. The rectal temperature was measured every 20 min in each mouse during the test. Unconditioned mice without VNL exposure (–) were analyzed as negative controls (black). PPA was used as an attractive-odor control (gray). Temperature differences before (T) and after (Tx) the transfer are compared. Error bars are SD (n = 3, 4, 4, 3, 3 animals). *p<0.05; ***p<0.005 (Student's t-test). VNL, vanillin; PPA, propionic acid.

The online version of this article includes the following source data for figure 6:

**Source data 1.** Odor detection of the PlxnC1 cKO in the habituation/dishabituation test.
**Source data 2.** Stress-induced hyperthermia test in the VNL-conditioned PlxnC1 cKO.

---

*Insel, 2010*). Furthermore, it has been reported that the KO of oxytocin (*Nishimori et al., 1996*) or its receptor (*Takayanagi et al., 2005*) affects social interactions and causes ASD-like responses. We, therefore, studied whether oxytocin is needed in neonates by the rescue experiment with the oxytocin KO (*Smith et al., 2019*).

To examine a possible effect of neonatal oxytocin on smooth social interactions later in life, we analyzed the oxytocin-administrated KO mice for social memory. In the oxytocin KO, increase in the sensitivity to the imprinted odor can still be seen (*Figure 7A*) and Sema7A expression is not affected by the KO (*Figure 7—figure supplement 1*). However, the positive quality is not imposed on the imprinted odor, as observed by loss of lasting interest (*Figure 7A*) and stress-reducing effect in the hyperthermia test (*Figure 7B*). In the social memory test, male mice demonstrate a characteristic decline in the time spent investigating an ovariectomized female during repeated pairings (*Ferguson et al., 2000*). In the male KO of oxytocin, time duration of sniffing did not decrease in the subsequent presentations of the same female mouse.

We then injected oxytocin into the KO pups by intraperitoneal administration (*Mizuno et al., 2015*) and tested their social memory after 10w. In the KO male treated with oxytocin at P0~6, sniffing times progressively decreased for the subsequent presentations (trials 1–5) of the same female mouse, but increased for a newly introduced female (trial 6) (*Figure 8A*). This rescue effect was not observed when the oxytocin was administrated after the critical period, at P8~14 (*Figure 8B*). Taken together, oxytocin in early neonates appears to be needed for smooth social interactions as adults by imposing positive quality on imprinted odor memory. Thus, imprinting (increase in the sensitivity) and positive memory formation are separately regulated by Sema7A/PlxnC1 signaling and oxytocin, respectively, during the neonatal critical period.

## Discussion

During development, an olfactory map is generated by a combination of A-P and D-V targeting using distinct sets of axon guidance molecules (*Nishizumi and Sakano, 2020*). These processes are independent from the neuronal activity of OSNs. The map is then refined by glomerular segregation molecules (*Serizawa et al., 2006*) whose expression is regulated by the intrinsic neuronal activity reported by *Reisert, 2010* (*Nakashima et al., 2019*). It appears that the intrinsic activity also mediates synapse formation within glomeruli. The olfactory map is further modified by the odor-evoked activity in neonates by enlarging the responsive glomeruli. This process is plastic and mediated by Sema7A/PlxnC1 signaling during the critical period. Activity-dependent Sema7A expression in OSNs triggers post-synaptic events in M/T cells via PlxnC1 (*Inoue et al., 2018*). Our present study demonstrates that Sema7A/PlxnC1 signaling is key to imprinting the neonatal odor experience during the critical period in mice. Rescue experiments with the Tg *Sema7a* gene in the CNG KO indicate that Sema7A is sufficient to promote activity-dependent synapse formation and dendrite selection within glomeruli. In M/T cells, PlxnC1 is localized to the dendrites only during the first week after birth.

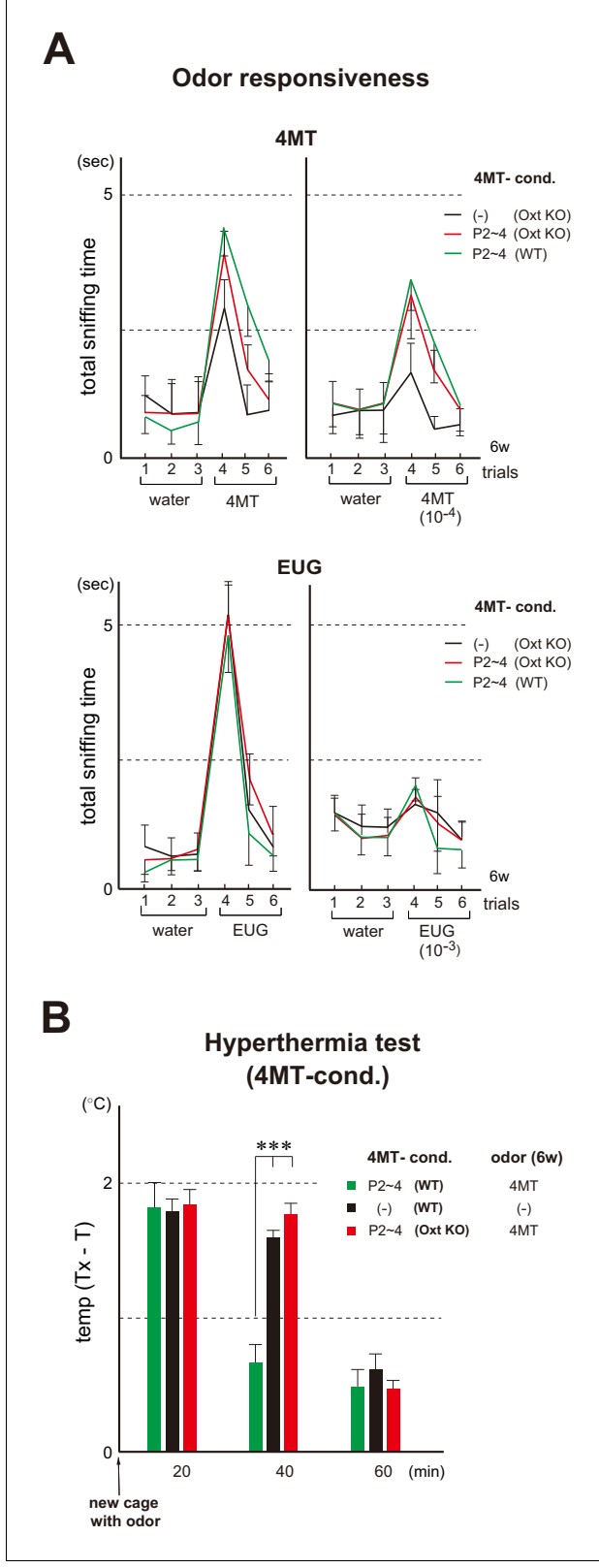

**Figure 7.** Odor imprinting in the oxytocin (Oxt) KO. (**A**) Odor responsiveness in the Oxt KO. Sniffing times (sec) were measured in the habituation/dishabituation test. The WT and Oxt KO were conditioned to 4MT at P2~4. Mice without conditioning (–) were analyzed as negative controls (n = 4 animals). Odorants used were as follows: 100 mM and 10 µM for 4MT (left); 6.4 M and 6.4 mM for EUG (right). Error bars are SD (n = 4, 5 animals). See also

*Figure 7 continued on next page*

*Figure 7 continued*

*Figure 1B* for the procedure. (B) Stress-induced hyperthermia test in the 4MT-conditioned Oxt KO. Pups of the WT (green) and Oxt KO (red) were exposed to 4MT at P2~4 and analyzed at 6w. Immediately after the transfer to a new cage, a filter paper spotted with 4MT was presented to the mice. The rectal temperature was measured every 20 min in each mouse during the test. Unconditioned mice without 4MT exposure (–) were analyzed as negative controls (black). Temperature differences before (T) and after (Tx) the transfer are compared. Error bars are SD (n = 4, 3, 4 animals). ***p<0.005 (Student's *t*-test). 4MT, 4-methyl-thiazole; EUG, eugenol.

The online version of this article includes the following source data and figure supplement(s) for figure 7:

**Source data 1.** Odor responsiveness in the Oxt KO.

**Source data 2.** Stress-induced hyperthermia test in the 4MT-conditioned Oxt KO.

**Figure supplement 1.** Expression of Sema7A in the Oxt KO.

Thus, Sema7A supports the activity-dependency of plastic changes within glomeruli and PlxnC1 limits the olfactory critical period to the early neonatal stage. It will be interesting if we can see shifts in the time window for the olfactory critical period by changing the time frame of PlxnC1 localization in the M/T-cell dendrites.

Sema7A expression increases in the responding glomeruli when newborns are exposed to a particular odorant. As a result, post-synaptic events are enhanced in the connecting M/T cells and dendrite selection is promoted. Unstimulated dendrites that connect other glomeruli are likely to be removed by synaptic competition. Thus, the number of primary dendrites increases in the stimulated glomerulus causing it to become larger. A specific set of enlarged glomeruli may be recognized as an imprinted odor code in adults. Sema7A/PlxnC1 signaling appears to have a prime role in adapting pups to environmental odorants and generating imprinted memory with lasting effects later in life. How does this imprinting occur in the context of circuit formation for adaptive odor responses? It has been reported that postnatal odor exposure heightens odor-evoked mitral-cell responses (*Liu and Urban, 2017*). It is likely that the enlarged glomeruli generate stronger signals of exposed odorants, which is then transmitted to the olfactory cortex, increasing the possibility that imprinted memory is established in neonates.

KO studies indicate that oxytocin is involved in imposing the positive quality on imprinted odor memory and is necessary for smooth social interactions. In order to determine when oxytocin is needed to achieve normal social skills, we attempted the rescue experiment for the oxytocin KO by intraperitoneal injection of oxytocin in neonates. Defective phenotypes of the KO mice were partly restored by this treatment when oxytocin was administrated during the critical period. The establishment of smooth social interactions by imprinted memory may be associated with a pleasant mental state in the nursed neonates. It is notable that in the oxytocin KO, Sema7A expression is not affected and the sensitivity to the imprinted odor still increases. Thus, imprinting and positive memory formation are independently regulated by Sema7A/PlxnC1 signaling and oxytocin, respectively, during the olfactory critical period. There are three major molecules involved in olfactory imprinting during the critical period. They are Sema7A which supports activity-dependency of the imprinting process, PlxnC1 which restricts the time frame of the critical period to the first week after birth, and oxytocin which is responsible for imposing the positive quality on imprinted memory.

It has been reported that early exposure to environmental odors affects social behavior later in life (*Logan et al., 2012*; *Mennella et al., 2001*; *Sullivan et al., 2000*; *Wilson and Sullivan, 1994*). Our present study suggests that odor-induced Sema7A/PlxnC1 signaling in neonates is essential for establishing odor imprinting, whose impairment cannot be recovered in adults. Male mice normally demonstrate strong curiosity toward unfamiliar mice of both genders. If Sema7A/PlxnC1 signaling is blocked during the critical period, the mice do not respond in their usual manner but demonstrate avoidance response to the stranger's scent. Correlating to this observation, there is a report showing that micro-deletion in the human chromosome 15q24 covering the *Sema7a* locus is associated with ASD (*McInnes et al., 2010*). It is well known that social stress induces the release of adrenocorticotropic hormone (*Backström and Winberg, 2013*). We assume that mice innately avoid stressful interactions with unfamiliar mice. However, imprinting mediated by Sema7A/PlxnC1 signaling may be necessary for the pups to adapt to their community. It will be interesting to study whether similar neonatal imprinting as that described in the mouse olfactory system can be found in humans, not

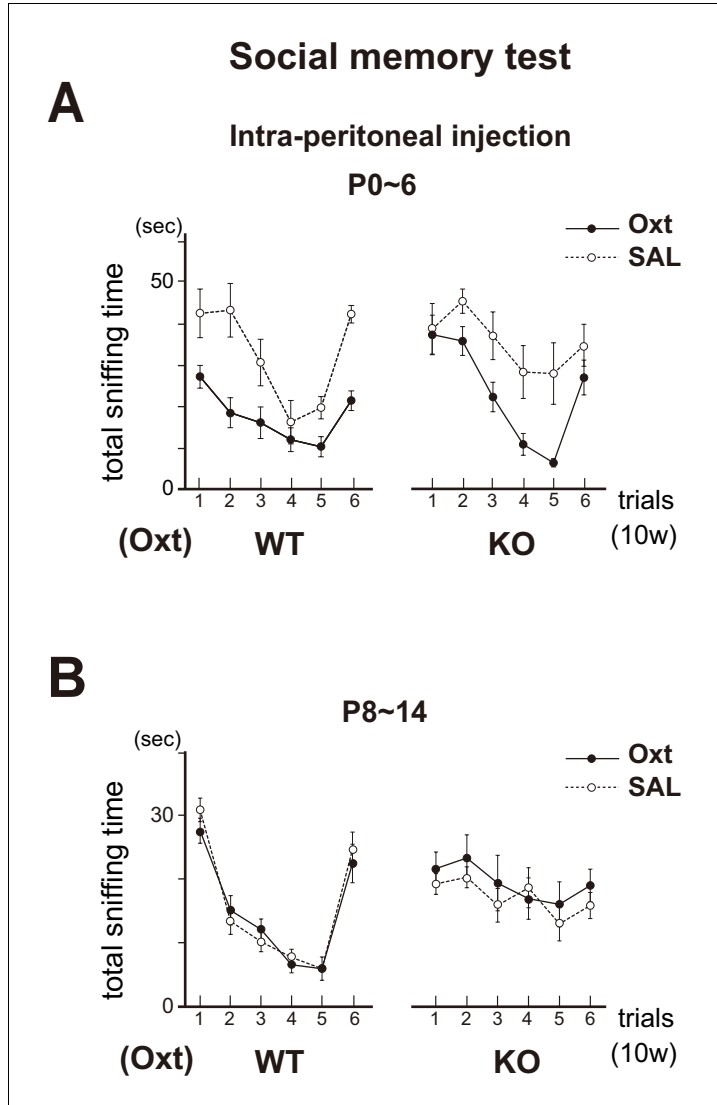

**Figure 8.** Social memory in the Oxt KO. The KO mice were administrated with oxytocin (Oxt) or saline (SAL) by intraperitoneal injection at P0~6 (**A**) or P8~14 (**B**). In the Oxt KO treated with SAL, time duration of sniffing did not change for the subsequent presentations of the same unfamiliar female mouse (trials 1–5). In contrast, in the KO treated with Oxt, sniffing time progressively decreased for the subsequent presentations of the same female (trials 1–5), but increased for the newly introduced female (trial 6), as found in the WT control. Data are shown as mean ± standard error. The male mice used were (P0~6, 8~14): WT-SAL (n = 6, 7), WT-Oxt (n = 8, 8), KO-SAL (n = 6, 9), and KO-Oxt group (n = 6, 9).

The online version of this article includes the following source data for figure 8:

**Source data 1.** Social memory in the Oxt KO.

only in the olfactory system, but also in other sensory systems. It is also important to determine when the critical period starts and how long it lasts in humans.

Our present study demonstrates that the imprinted odor induces positive responses even when its quality is innately aversive. In such a situation, where the hard-wired decision is modified by imprinting to elicit the final response, how does the amygdala arbitrate the conflicting decisions, one which is innate and the other which is modified by imprinted memory? In order to reconcile the two decisions, there may be a cross talk between the positive and negative valence cells in the amygdala (*Mori and Sakano, 2021*). These studies will give new insights into our understanding of decision making in humans and will shed light on the neurodevelopmental disorders, such as ASD and attachment disorders, that may be caused by improper sensory inputs during the critical period.

# Materials and methods

**Key resources table**

| Reagent type (species) or resource | Designation | Source or reference | Identifiers | Additional information |
|---|---|---|---|---|
| Genetic reagent (*M. musculus*) | *Cnga2* KO | Jackson Laboratory | Stock #: 002905 RRID:MGI:3717661 | PMID:15071119 |
| Genetic reagent (*M. musculus*) | *Sema7a* KO | Jackson Laboratory | Stock #: 005128 RRID:MGI:2683896 | PMID:12879062 |
| Genetic reagent (*M. musculus*) | *Oxytocin* KO | PMID:8876199 | RRID:MGI:3603795 | Dr. Katsuhiko Nishimori (Tohoku University) |
| Genetic reagent (*M. musculus*) | *BAC Olfr1510/1511 Tg* | PMID:21105914 | | Dr. Hitoshi Sakano (University of Fukui) |
| Genetic reagent (*M. musculus*) | *pOlfr16- Lofr226* | PMID:16990513 | | RIKEN BRC (RBRC02931) |
| Genetic reagent (*M. musculus*) | *pOlfr16- Lofr226 (RDY)* | PMID:16990513 | | RIKEN BRC (RBRC02933) |
| Genetic reagent (*M. musculus*) | *Plxnc1$^{flox/flox}$* | PMID:29743476 | | RIKEN BDR (Acc. #: CDB0908K) |
| Genetic reagent (*M. musculus*) | *pOlfr16- Lofr226-Sema7a* | This paper | | Dr. Hitoshi Sakano (University of Fukui) |
| Genetic reagent (*M. musculus*) | *pOlfr16- Lofr226-Sema7a (Y213S)* | This paper | | Dr. Hitoshi Sakano (University of Fukui) |
| Genetic reagent (*M. musculus*) | *Tg(Pcdh21-cre)BYoko* | PMID:16106355 | RRID:MGI:4940883 | RIKEN BRC (RBRC02189) |
| antibody | anti-Sema7A | R and D Systems | Cat. #: AF-1835 | IF(1:3000) |
| antibody | anti-PlxnC1 | Abcam | discontinued | IF(1:3000) |
| antibody | anti-CNG-A2 | Alomone Labs | Cat. #:APC-045 | IF(1:200) |
| antibody | anti-vGlut2 | Millipore | Cat. #: AB2251-l | IF(1:1000) |
| antibody | anti-GluR1 | Abcam | Cat. #: ab51092 | IF(1:1000) |
| antibody | anti-GFP | Thermo Fisher Scientific | Cat. #: A-10260 | IF(1:1000) |
| antibody | anti-Lucifer yellow | Thermo Fisher Scientific | Cat. #: A-5750 | IF(1:2000) |
| antibody | anti-EGR1 | Abcam | Cat. #: ab6054 | IF(1:1000) |

## Contact for reagent and resource sharing

Further information and requests for resources and reagents should be directed to and will be fulfilled by the lead contact, Hitoshi Sakano (sakano.hts@gmail.com).

## Mutant mice

The *Cnga2* KO (No. 002905) (*Lin et al., 2000*) and *Sema7a* KO (No. 005128) (*Pasterkamp et al., 2003*) were purchased from the Jackson Laboratory. The KO mice for oxytocin (*Nishimori et al., 1996*) was obtained from Tohoku University. BAC Tg mice containing the *MOR29A* (also known as *Olfr1510*) and *MOR29B* (also known as *Olfr1511*) genes (*Tsuboi et al., 2011*), Tg mice expressing the *rI7* (also known as *Lofr226*) gene with the *MOR23* (also known as *Olfr16*) promoter (*Imai et al., 2006*), and *rI7 (RDY)-ires-gap-YFP* mice (*Imai et al., 2006*) were generated in our group. For the cell-type-specific cKOs of *Plxnc1*, we cloned the exon 5 of *Plxnc1* into the double-floxed, inverted open-reading frame plasmid, *DT-A/Conditional FW* (http://www2.clst.riken.jp/arg/cassettes.html). The cKOs mice were generated according to the published protocol (http://www2.clst.riken.jp/arg/gene_targeting.html). To generate the Tg mice carrying the *rI7-ires-Sema7a-ires-YFP* or the *rI7-ires-Sema7a (Y213S)-ires-YFP*, an *Eag*I cassette containing the transgene was inserted into the *Eag*I site in the *pMOR23-rI7-ires-gap-YFP* construct. All five transgenic lines obtained for each, successfully expressed Tg-Sema7A in the rI7-positive OSNs, although expression levels of Tg-Sema7A were ~10% compared with the endogenous Sema7A. The *Pcdh21-Cre* mouse (*Nagai et al., 2005*)

was obtained from the RIKEN Bioresource Research Center (RBRC02189) and crossed with the $Plxnc1^{flox/flox}$ mice.

All animals were housed under a standard 12 hr light/dark cycle (light on from 8 am to 8 pm), constant temperature (22 ± 2℃), and humidity (50 ± 10%). Food and water were provided ad libitum. All animal experiments were approved by the Animal Care Committees at University of Fukui and Azabu University.

## Naris occlusion

Unilateral naris occlusion was performed at P0 on the right nostril with an electrical cautery, SURE SX-10C (Ishizaki, Japan). The occluded pups were examined daily to ensure blockade of the occluded nostril by checking the scar formation. The right-side naris was reopened at various time points during the postnatal period using a 31G needle. In some experiments, occlusion was initiated after P0 for different time lengths during the neonatal period.

## Antibodies

Antibodies used in this study are as follows: Sema7A (goat, 1:3000, #AF-1835, R and D Systems); PlxnC1 (goat, 1:3000, discontinued, Abcam); CNG-A2 (rabbit, 1:200, #APC-045, Alomone Labs); vGlut2 (guinea pig, 1:1000, #AB2251-l, Millipore); GluR1 (rabbit, 1:1000, #ab51092, Abcam); GFP (rabbit, 1:1000, #A-10260, Thermo Fisher Scientific); Lucifer yellow (rabbit, 1:2000, #A-5750, Thermo Fisher Scientific); and EGR1 (rabbit, 1:1000, #ab6054, Abcam).

## Immunostaining

For immunohistochemistry, OB sections were fixed with 4% paraformaldehyde in PBS and treated with the primary and secondary antibodies. The sections were then photographed with a fluorescence microscope, Model IX70 (Olympus), coupled to a cooled CCD camera, C4742-95-12ERG (Hamamatsu Photonics).

## Intensity measurement

For fluorescent signals of immunostaining, digital images were taken with a digital CCD camera, C4742-95-12ERG (Hamamatsu Photonics), or with two-photon microscopy (Olympus FV1000). Tone was reversed and a monochrome image was used for the measurement. For staining of the OB sections, digital images were taken with a digital CCD camera, Model DP70 (Olympus). To quantify the staining level of each glomerulus, the mean pixel intensity within the region surrounded by the peri-glomerular-cell nuclei was measured using Scion Image (Scion Corp.).

## Neonatal odor conditioning

Before conditioning, mother mice were pre-exposed to the odor for 10 min, three times a day for 3 consecutive days. A filter paper (2×2 cm) spotted with a test odorant was placed near the mice. For conditioning, pups were exposed to the odor for 10 min with the foster mother, three times a day for 3 consecutive days. Odor concentration was 0.5 µl of 20 mM for vanillin (VNL) or 100 mM for 4-methyl-thiazole (4MT) (Wako).

## Habituation/dishabituation test

For the test, adult littermates (6w-old male) were kept individually in a new cage (26×40×18 cm) containing a plain filter paper for 5 min prior to the experiment. This treatment was to habituate the mice to a filter paper and to evaluate the net curiosity to a novel odor in the following filter presentation. During the treatment, about 8% of total mice did not demonstrate any interest in the plain filter paper. We precluded such mice for further experiments. In the test, a filter paper spotted with 0.5 µl of distilled water was intermittently presented for 3 min. This operation was repeated three times with 1 min intervals. Next, a filter paper spotted with the 1st test odorant was presented three times as in the presentation of water-spotted filter paper (habituation). Then, the 2nd test odorant was presented three times in the same way (dishabituation). Investigation times (s) observed during each odorant presentation were measured. In this test, the mice were used only once not to confound the data due to the previous learning. The test was blinded to avoid unconscious bias and to insure the objectivity. The mouse behavior was recorded with a digital video camera, DCR-SR100

(SONY). In this assay, investigation was defined as a nasal contact with filter papers within 1 mm of distance. Odorants used were (+)-carvone (CAR) (Sigma-Aldrich), (–)-CAR (Sigma-Aldrich), eugenol (EUG) (Tokyo-Kasei), and vanillin (VNL) (Wako).

## Stress-induced hyperthermia test

For the stress-induced hyperthermia test (Spooren et al., 2002), mice (6w-old) were individually habituated to the cage and their body temperature was checked to be normal. The mice were then transferred to a new cage and exposed to a testing odorant. The rectal temperature was measured with a digital thermistor, BAT-12 (Physitemp Instruments Inc) by inserting the probe to a length of 10–20 mm until stable reading was obtained. Differences of rectal temperatures before (T) and after (Tx) the transfer were individually measured every 20 min. Odor concentrations were 0.5 µl of 20 mM for vanillin (VNL), 100 mM for propionic acid (PPA) (Tokyo-Kasei), and 100 mM for 4-methyl-thiazole (4MT) (Wako).

## In situ hybridization

To prepare cRNA probes, DNA fragments of 500–3,000 bp were amplified by PCR from the OB cDNA of C57BL/6 mice. Only unique sequences were amplified for each gene. PCR products were subcloned into the pGEM-T vector (Promega) and used as templates to make cRNA probes. Hybridization was performed according to the standard method (Inoue et al., 2018).

## Electron microscopy

Mice were transcardially perfused with 2% paraformaldehyde (PFA) and 2.5% glutaraldehyde (GA) in 0.1 M PB. The mice were then immersed in the same buffer at 4℃ for 1–2 hr and rinsed in 0.1 M PB prior to being sectioned transversely into 50 µm slices with a vibratome, LinearSlicer PRO7 (D.S.K.). The sections for electron microscopy were incubated in 2% osmium tetroxide for 60 min, dehydrated through an ascending ethanol series, and embedded in EPON. Thin sections (70–100 nm) were then sliced with EM UC7 (Leica), mounted on Cu grids, and examined in a transmission electron microscope VE-9800 (KEYENCE). Images were photographed at primary magnification.

## Odor preference test

For the social response test, 6w-old adult littermates (male mice) were individually habituated in a new cage (26×40×18 cm). Then, food pellet (Japan SLC, Inc) or beds with unfamiliar mouse odor were carefully placed not to disturb the mice in a corner of the test cage. To avoid direct contacts of the mouse nose, the food pellet or bed samples were presented in a small plastic cup. The cup positions were alternated to avoid side preference. This apparatus evaluates a simple social interest in the volatile smell of unfamiliar mice, excluding visual, auditory, and vomeronasal inputs. Times (s) spent in the room with or without a test odorant were measured during the 5-min test period. The mice were used only once to avoid possible bias attribute to learning. The test was blinded to avoid unconscious bias to insure the objectivity. The mouse behavior was recorded with a digital video camera, DCR-SR100 (SONY).

## Three-chamber social interaction test

Littermate mice (6w-old male) were used in the experiment. The three-chamber test was conducted as described by Moy et al., 2004. Before the test, a mouse was allowed to explore all three chambers divided with plastic walls with a loophole in the apparatus for 10 min. Only those mice moving actively in the cage were used for the test. Then, the test mouse was placed in the center chamber, an empty plastic cage was to the right, and an unfamiliar mouse (male C57BL/6J) in a cage was to the left. The mouse being tested was allowed to move freely and explore all three chambers for 15 min. The mouse behavior was recorded with a digital video camera, DCR-SR100 (SONY). Time duration spent in each room was measured during the 15-min test period. The positions of the empty cage and unfamiliar mouse were alternated to avoid left/right side preference.

## Social memory test

All pups in each litter received either an intra-peritoneal injection of oxytocin (2 µg/20 µl for P0~6 and 5 µg/20 µl for P8~14) or saline (SAL, 20 µl) on postnatal days 0, 2, 4, and 6, or 8, 10, 12, and 14.

The same cohort of male mice that were examined for alloparental responsiveness, was tested by the habituation/dishabituation paradigm (*Ferguson et al., 2000* and *Ferguson et al., 2001*). In this paradigm, mice exposed to the same stimulus animal should decrease their investigation times with each exposure (habituation) and increase their investigation time when a novel animal is introduced (dishabituation). One male subject was transferred from its group to an individual cage of the same size as the breeding cage for 7–10 days before testing for the establishment of a home-cage territory. Testing began with a stimulus mouse, an ovariectomized female unfamiliar to the subject, was introduced into the home cage of each subject for a 1 min confrontation. At the end of the 1 min trial, the stimulus was removed and returned to its home cage. This trial was repeated for five times with 10 min intertrial intervals, and each stimulus was introduced to the same male subject in all five trials. In the 6th dishabituation trial, an ovariectomized female mouse different from the previous one was introduced into the home cage of the subject mouse. Each trial was recorded with a CCD camera connected to a video recorder. The duration of sniffing the stimulus' anogenital area was measured by a well-trained observer who was blinded to both the mice genotypes and treatments.

## Statistical analyses

All statistical analyses were performed using Excel 2016 (Microsoft) with the Statcel2 add-on (OMS).

## Acknowledgements

We are grateful to H Naritsuka for her advice in single-cell labeling and H Sagara for his help in EM analysis. We thank members of our laboratory for valuable suggestions and discussion. This research was supported by the JSPS KAKENHI (Grant Numbers 24000014 and 17H06160 to HS; 17K19386 and 20K06909 to HN) and the MEXT KAKENHI (Grant Numbers 17H05943 and 19H04745 to HN). This work was also supported by the Japan Foundation for Applied Enzymology (J180000322 to NI) and the Naito Foundation (HN).

## Additional information

### Funding

| Funder | Grant reference number | Author |
| --- | --- | --- |
| JSPS | KAKENHI 24000014&17H06160 | Hitoshi Sakano |
| JSPS | KAKENHI 17K19386 & 20K06909 | Hirofumi Nishizumi |
| Japan Foundation for Applied Enzymology | J180000322 | Nobuko Inoue |
| The Naito Foundation | | Hirofumi Nishizumi |
| MEXT | KAKENHI 17H05943 & 19H04745 | Hirofumi Nishizumi |

The funders had no role in study design, data collection and interpretation, or the decision to submit the work for publication.

### Author contributions

Nobuko Inoue, Conceptualization, Resources, Data curation, Software, Formal analysis, Funding acquisition, Validation, Investigation, Visualization, Methodology, Writing - original draft, Writing - review and editing; Hirofumi Nishizumi, Conceptualization, Formal analysis, Supervision, Funding acquisition, Validation, Investigation, Visualization, Writing - original draft, Writing - review and editing; Rumi Ooyama, Data curation, Investigation; Kazutaka Mogi, Data curation, Formal analysis, Validation, Investigation; Katsuhiko Nishimori, Resources; Takefumi Kikusui, Data curation, Supervision, Validation, Writing - review and editing; Hitoshi Sakano, Conceptualization, Supervision, Funding acquisition, Validation, Writing - original draft, Project administration, Writing - review and editing

## Author ORCIDs

Hirofumi Nishizumi (iD) https://orcid.org/0000-0002-6083-9233
Takefumi Kikusui (iD) http://orcid.org/0000-0002-3134-5859
Hitoshi Sakano (iD) https://orcid.org/0000-0002-5333-1722

## Ethics

Animal experimentation: the Animal Care Committees at University of Fukui and Azabu University.

## Decision letter and Author response

Decision letter https://doi.org/10.7554/eLife.65078.sa1
Author response https://doi.org/10.7554/eLife.65078.sa2

## Additional files

### Supplementary files

• Transparent reporting form

### Data availability

All data generated or analyzed during this study are included in the manuscript and supporting files.

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
