## [Decision Letter]

**Acceptance summary:**

This manuscript investigates a role for Sema7A/PlxnC1 in an olfactory critical period. The authors observed that mice who encounter odors as neonates (before ~P7) display altered behavioral and neuronal responses to those odors as adults. Neonatal-experienced odors are investigated more vigorously, and mildly reduce stress, perhaps akin to familiar nest odors. Disruption of Plxn1 alters neuronal circuitry and eliminates the behavioral effects of neonatal odor exposure.

**Decision letter after peer review:**

Thank you for submitting your article "The Olfactory Critical Period is Defined by Activity-Dependent Synapse Formation Induced by Sema7A/PlxnC1 Signaling" for consideration by *eLife*. Your article has been reviewed by three peer reviewers, one of whom is a member of our Board of Reviewing Editors, and the evaluation has been overseen by Gary Westbrook as the Senior Editor. The reviewers have opted to remain anonymous. The reviewers have discussed the reviews with one another and the Reviewing Editor has drafted this decision to help you prepare a revised submission.

Summary:

Sensory systems develop with “critical periods” where sensory experience causes long-lasting structural changes within the brain with lasting impact on adult behavior. Here, the authors identify key development factors (Sema7A/PlxnC1) that are required for an olfactory critical period in mice, and loss of these factors, or the hormone oxytocin, impairs olfactory learning. The reviewers all agreed that the manuscript presented important findings about the roles of Sema7a and PlxnC1 in odor imprinting that will be of broad interest. However, there were also substantial concerns from all reviewers that results were not all coherent and thematically connected. Additional experiments are required, as detailed below in Essential revisions, to justify conclusions drawn from the data.

Essential revisions:

1) All reviewers agree that a link between oxytocin and Sema7a/PlxnC1 is lacking. Do the authors observe alterations in Sema7a signaling in oxytocin knockout mice? If so, the data would strengthen the manuscript considerably; if not, it is advised that the paper could be strengthened by removing the disconnected oxytocin data.

2) The reviewers request testing Sema7a mutants for behavioral effects in odor paradigms presented (including those in Figure 2 and Figure 6).

3) The authors should improve description and validation of several mouse models.

Reviewer #1:

This manuscript investigates a role for Sema7A/PlxnC1 in an olfactory critical period. The authors observe that mice who encounter odors as neonates (before ~P7) display altered behavioral and neuronal responses to those odors as adults. Neonatal-experienced odors are investigated more vigorously, and mildly induce stress, perhaps akin to familiar nest odors. The authors report that neonatal odor exposure increases Sema7a expression, enlarges target glomeruli through increased numbers of periglomerular cells, and increases the percentage of mature mitral cells. The authors then show that knockout of Plxn1 or oxytocin impairs social odor recognition. There are many interesting findings here, and the manuscript is well written. Yet, some of the key findings are not thematically connected and additional experiments are needed to strengthen links between Sema7A signaling, oxytocin, and behavioral responses.

1) The authors show that neonatal-experienced odors are more attractive to odors, and that neonatal odors upregulate Sema7A in responsive neurons. Yet, a role for Sema7A/PlxnC1 in enhanced neonatal odor investigation is not reported. Do Sema7a knockout mice lose effects of neonatal-experienced odors described in Figure 2? Are they rescued by re-expression of Sema7A in olfactory sensory neurons (using Tg-Sema7a)?

2) The connection between oxytocin and Sema7A signaling is unclear. Does oxytocin alter expression/localization of PlxnC1 and/or Sema7A? Can the authors propose and test models for how oxytocin impacts olfactory circuits?

3) As the authors have demonstrated beautifully in prior studies, activity through Gs is needed for map establishment while Golf/CNGa2 activity is needed for odor-evoked activity and maintenance. The paper would benefit by further discussion of the roles of “activity” and that whether Sema7a contributes only to maintenance and not map establishment.

4) Provide additional explanation in the main text and validation of the Tg-Sema7A line. What % of olfactory sensory neurons re-express Sema7a? Likewise, additional validation of the mitral cell-specific PlxnC1 knockout is needed.

5) Representative images should be shown for all data presented as a signal intensity, including Figures 1A, 3A, B, 4, and 5. Also, images showing the differences between mature/immature dendrite selection are needed (3B).

6) A home cage nest odor would be a nice positive control for Figure 2B, for context to understand the magnitude of the observed behavioral response.

7) The authors claim (Results) that both tufted and periglomerular cells increase in number, but Figure 3C only provides supporting data for periglomerular cells.

Reviewer #2:

1) It is not clear exactly what the relationship is between the oxytocin experiments and the rest of this study. First, do oxytocin mutants show alterations Sema7a expression, or PlexC1 temporal distribution in the tuft structure? Why is a different behavioral assay used for the oxytocin experiments as compared to the social interaction assay used in the Plexc1 conditional mutants.

Overall, there remain questions regarding the behavioral experiments such that if the oxytocin experiment was not part of this study, it would not in my view reduce interest or impact of the work.

2) Does loss of Sema7a in ORs lead to the same behavioral phenotypes observed in the cKO M/T PlexC1 mutants? And is there any way to show that the loss-of-function (LOF) experiment involving PlexC1 is dependent upon LOF during the critical period? For this current study, addressing the Sema7a requirement genetically in this behavioral assay seems to be an issue that should be tackled in the context of the social interaction assay, if it is to be presented here.

3) I do not have many comments on the first part of the study, however, and find it to be quite interesting. I do wonder:

a) If in the conditional CNG mutants, whether or not there is any change in the temporal localization of PlexC1 with respect to the presented postnatal localization in the tuft structure-is this, too, activity dependent and/or regulated by Sema7a during the critical period.

b) Is there any evidence that Sema7a OR expression normally follows a time course that directly links it to the critical period. This seems like a point that could be easily addressed.

Reviewer #3:

1) The conclusion of the paper appears to be based on the following logical connection: activity driven changes in synaptic function in the olfactory bulb is restricted to the first week after birth because activities drive the expression of Sem7A expression, which interacts with PlxnC1, expressed only during the critical period. This is not explicitly stated, nor extensively discussed. Although the model is very attractive, the experimental evidence is correlational. The conclusion that Sema7A/PlxnC1 support the olfactory critical period is largely based on the restricted dendritic expression PlxnC1 during the first postnatal week.

2) Naris occlusion and odor exposure experiments are complementary to each other to provide support for activity dependent regulation of Sema7A expression, but there is a discrepancy in the results. Naris occlusion experiment show that the level of synaptic proteins is normal if occlusion is reopened before P7. However, odor exposure using vanillin between P2 and P4 altered odor habituation and stress response. The authors have not explained the discrepancy in the experimental design and results with regard of the difference in timing.

3) The CNGA2 KO experiments lend support for activity-dependent expression of Sema7A, and the Sema7A rescue experiment provides further support that Sema7A could induced synaptic protein expression. However, different glomeruli exhibit different levels of Sema7A expression. Is this difference intrinsic to different types of OSNs, or is it caused by differences in neural activity?

4) The conditional PlxnC1 KO mice need more rigorous testing. If activity-dependent expression of Sema7A and Sema7A/PlxnC1 interaction are important for synapse formation in the olfactory glomeruli, odor detection and discrimination are expected to be affected. No experiments have been conducted to address this prediction. It is also not known whether the KO abolishes the phenotypes observed in Figure 2, which are associated with vanillin exposure during the critical period. Moreover, it is not clearly stated how the three-chamber assay is related olfactory imprinting. A likely explanation of the observed impairment of odor preference and social behavior is that the overall olfactory sense has been compromised.

5) The oxytocin KO experiment is interesting. However, the behavioral phenotype is difficult to interpret. WT mice exhibit a decline in investigation of an ovariectomized female whereas the oxy-/- mice show persistent investigation. The authors interpret the decline as an indication of social memory, but the lack of decline/persistent investigation in the oxy-/- mice could be interpreted as prosocial interactions. This interpretation would be consistent with the decrease of investigation in WT mice after P0-P6 oxytocin injection. Importantly, the authors have not addressed how this experiment is related to activity-dependent modulation of glomerular connectivity.

6) The authors present data with quantification, but primary data (images) are not presented. This makes it difficult to assess the accuracy of the quantification. Specifically, primary images demonstrating the selective induction of Sema7A expression in the MOR29A glomeruli after vanillin exposure should be compared with no-odor control side-by-side.

---

## [Author Response]

Essential revisions:1) All reviewers agree that a link between oxytocin and Sema7a/PlxnC1 is lacking. Do the authors observe alterations in Sema7a signaling in oxytocin knockout mice? If so, the data would strengthen the manuscript considerably; if not, it is advised that the paper could be strengthened by removing the disconnected oxytocin data.

The editor points out that a link between oxytocin and Sema7A/PlxnC1 is lacking, which is a theme raised by all the reviewers. Our description was insufficient and we should have provided a better explanation in our manuscript.

We continued our investigation into oxytocin, after the completion of PlxnC1 cKO experiments. This is because two of the three observations we found were behaviors that could not solely be explained at the neuronal level influenced by Sema7A/PlxnC1 signaling. In the PlxnC1 cKOs, the normal behavioral response to odors, which the mice were previously exposed to during the critical period, (1) increased sensitivity, (2) lasting interest, and (3) easing of stress were all abolished. Increased sensitivity is a form of imprinted memory, and its blockage can be explained by the failure of synapse formation within the glomeruli in the PlxnC1 cKO. However, lasting interest and easing of stress are examples of positive memories that need to be further formed in the central brain – a neuronal level not affected by the PlxnC1 cKO.

We therefore, turned our attention to various neuronal hormones that may influence the formation of conditioned memory, and homed-in on oxytocin. In the oxytocin KO studies, we observed that the two conditioned memories, lasting interest and easing of stress, were both perturbed, but increased sensitivity was not. This indicated that Sema7A/PlxnC1 signaling is necessary for odor imprinting, reflected by the formation of all three learning responses. However, oxytocin is then necessary for the further establishment of positive memory (lasting interest and easing of stress).

To the question of whether the authors observed alterations in Sema7A signaling in oxytocin knockout mice: Sema7A expression is not affected in the oxytocin KO (Figure 7—figure supplement 1). In the KO, imprinting still occurs as evidenced by an increase in odor sensitivity (Figure 7A), but conditioned learning does not. In other words, positive quality is not imposed on the imprinted odor, as observed by a negative response to the hyperthermia test. New data for the stress-reducing experiment is now shown in Figure 7B. We conclude that imprinting and conditioned memory formation are independently regulated by Sema7A/PlxnC1 and oxytocin, respectively, during the critical period. These observations also demonstrate that Sema7A/PlxnC1 signaling is not downstream of oxytocin signaling, although oxytocin is needed for establishing positive memory of imprinted odors.

These are now described in the revised text with new figures (Figure 7 and Figure 7—figure supplement 1). As both signaling systems play essential roles in establishing imprinted memory during the critical period, we would like to keep the description of oxytocin in the manuscript.

2) The reviewers request testing Sema7a mutants for behavioral effects in odor paradigms presented (including those in Figure 2 and Figure 6).

The editor makes mention of behavioral effects in the Sema7A KO. At present, we only have a total KO line for Sema7A, but not its cKO specific to OSNs. The total KO of Sema7A would not be appropriate for behavioral tests, because Sema7A is expressed in various brain regions (Carcea et al., 2014). Fortunately, however, we have the cKO of PlxnC1 specific to M/T cells, which blocks Sema7A signaling in the neonatal glomeruli. We have previously reported that the effects of Sema7A total KO and M/T-cell-specific PlxnC1 cKO are essentially the same in synapse formation and dendrite selection within glomeruli (Inoue et al., 2018).

We analyzed the PlxnC1 cKO mice for their responses to the conditioned odor in the habituation/dishabituation test and hyperthermia test. In the PlxnC1 cKO, neither an increase in the sensitivity, nor interest toward the conditioned odor was observed (Figure 6A). Stress was not eased by the conditioned odor in the hyperthermia test (Figure 6D). For their social responses, results in the three-chamber test are already shown in Figure 6C, using the PlxnC1 cKO where the sema7A signaling is blocked in M/T cells. In the cKO, abnormal social behavior is observed, avoiding social interactions with unfamiliar mice.

These are now described in the text with new figures (Figure 6A and D).

3) The authors should improve description and validation of several mouse models.

As advised by the editor, we described additional information for the mouse lines for Tg-Sema7A (Figure 5—figure supplement 2A) and PlxnC1 cKO in the text (Results and Materials and methods).

Reviewer #1:This manuscript investigates a role for Sema7A/PlxnC1 in an olfactory critical period. The authors observe that mice who encounter odors as neonates (before ~P7) display altered behavioral and neuronal responses to those odors as adults. Neonatal-experienced odors are investigated more vigorously, and mildly induce stress, perhaps akin to familiar nest odors. The authors report that neonatal odor exposure increases Sema7a expression, enlarges target glomeruli through increased numbers of periglomerular cells, and increases the percentage of mature mitral cells. The authors then show that knockout of Plxn1 or oxytocin impairs social odor recognition. There are many interesting findings here, and the manuscript is well written. Yet, some of the key findings are not thematically connected and additional experiments are needed to strengthen links between Sema7A signaling, oxytocin, and behavioral responses.1) The authors show that neonatal-experienced odors are more attractive to odors, and that neonatal odors upregulate Sema7A in responsive neurons. Yet, a role for Sema7A/PlxnC1 in enhanced neonatal odor investigation is not reported. Do Sema7a knockout mice lose effects of neonatal-experienced odors described in Figure 2? Are they rescued by re-expression of Sema7A in olfactory sensory neurons (using Tg-Sema7a)?

The mice demonstrate positive responses to the conditioned odor as adults. Stress is lowered by the imprinted odor in the hyperthermia test. This stress-easing effect cannot be seen when Sema7A signaling is blocked in the cKO of PlxnC1 (Figure 6D). A role of Sema7A/PlxnC1 signaling in odor imprinting is now discussed more in the text with new behavioral data of the PlxnC1 cKO (Figure 6A and D).

The reviewer also suggests a rescue experiment in the Sema7A KO using the Tg-Sema7A. This would be an excellent experiment to strengthen the Sema7A data. However, the Tg-Sema7A described in Figure 5 can be analyzed only in the rI7-expressing OSNs, because of its *MOR23* promoter for co-expression. Furthermore, our Sema7A KO is a total KO and unfortunately would not be appropriate for behavioral analyses because Sema7A is expressed in various brain areas (Carcea et al., 2014). For these reasons, the rescue experiment advised by the reviewer cannot be performed without generating a new cKO line.

2) The connection between oxytocin and Sema7A signaling is unclear. Does oxytocin alter expression/localization of PlxnC1 and/or Sema7A? Can the authors propose and test models for how oxytocin impacts olfactory circuits?

As also pointed out by the editor and other reviewers, we agree that a link between the two signaling systems, Sema7A/PlxnC1 and oxytocin is not clearly described in the manuscript. With the addition of new data, we hope that this point is made clearer in our revised manuscript.

As mentioned in the responses to other reviewers/editor, these two signaling systems are not directly linked to one another in imprinting. Separately from Sema7A/PlxnC1 signaling, oxytocin in neonates is needed for imposing the positive quality on neonatal odor memory and positive responses to the imprinted odor is not observed in the oxytocin KO (Figure 7B). Interestingly, however, increase in the sensitivity to the experienced odor can still be seen in the oxytocin KO (Figure 7A). Furthermore, Sema7A expression is not affected in the oxytocin KO (Figure 7—figure supplement 1). Our observations demonstrate that changes in the odor sensitivity and odor quality are separately regulated by Sema7A/PlxnC1 and oxytocin, respectively. These are now described in the text with new data (Figure 7) and discussed more in the text.

3) As the authors have demonstrated beautifully in prior studies, activity through Gs is needed for map establishment while Golf/CNGa2 activity is needed for odor-evoked activity and maintenance. The paper would benefit by further discussion of the roles of “activity” and that whether Sema7a contributes only to maintenance and not map establishment.

We agree that this is an important point. The basic architecture of the olfactory circuit is generated by the genetic program, then further refined and modified by neuronal activity. During development, a coarse olfactory map is generated by the combination of anterior/posterior and dorsal/ventral targeting using separate sets of axon guidance molecules (Sakano, 2010). These processes are independent from the neuronal activity of OSNs. The map is then refined by axon-sorting molecules whose expression is regulated by neuronal activity of OSNs. It has been reported that this activity is not odor-evoked, but is caused by the spontaneous firing of OSNs (Reisert, 2010； Nakashima et al., 2019). The intrinsic activity of OSNs also mediates basic synapse formation within glomeruli in the absence of environmental odorants. The olfactory map is further modified by the odor-evoked neuronal activity of OSNs by enlarging the responding glomeruli. This process is plastic and mediated by Sema7A/PlxnC1 signaling during the neonatal critical period. Sema7A expression is induced by OR-specific activity and promotes post-synaptic events within glomeruli (Inoue et al., 2018). Although the blockage of Sema7A/PlxnC1 signaling diminishes odor-evoked synapse formation, targeting of OSN axons to the OB is not affected. These are now described in the Introduction and Discussion.

4) Provide additional explanation in the main text and validation of the Tg-Sema7A line. What % of olfactory sensory neurons re-express Sema7a? Likewise, additional validation of the mitral cell-specific PlxnC1 knockout is needed.

For the Tg-Sema7A mice, we obtained five transgenic lines, all of which successfully expressed Tg-Sema7A in the rI7-positive glomeruli. Compared with the endogenous Sema7A in the WT, expression levels of Tg-Sema7A were ~10% within the rI7 glomeruli in the CNG/Sema7A KO (Results) (Figure 5—figure supplement 2A). Since the staining signals of Sema7A were not strong enough within individual cells for counting, exact % of OSNs re-expressing Sema7A were difficult to determine (Materials and methods).

As for the M/T-specific cKO of PlxnC1, it was generated using the Pcdh21-Cre mouse as a KO driver (Inoue et al., 2018). Absence of PlxnC1 expression was confirmed by in situ hybridization and immunostaining. These are now mentioned in the text (Results and Materials and methods).

5) Representative images should be shown for all data presented as a signal intensity, including Figures 1A, 3A, B, 4, and 5. Also, images showing the differences between mature/immature dendrite selection are needed (3B).

As advised, representative images are now shown in the new supplementary figures as follows:

Figure 1—figure supplement 1 is for the data in Figure 1A.

Figure 3—figure supplement 1A is for Figure 3A.

Figure 3—figure supplement 1B and Figure 3—figure supplement 2 are for Figure 3B.

Figure 3—figure supplement 1A and Figure 4—figure supplement 1 are for Figure 4.

Figure 5—figure supplement 2 is for Figure 5.

6) A home cage nest odor would be a nice positive control for Figure 2B, for context to understand the magnitude of the observed behavioral response.

As the reviewer advised, home-cage nest would be a nice positive control. Although we did not use it in our hyperthermia experiment, we tested propionic acid (PPA) as a positive control, whose odor quality is innately attractive (Figure 6D). PPA lowers stress, but its stress-reducing effect is more moderate compared with the imprinted VNL odor whose quality is innately neutral. This is now described in the text.

7) The authors claim (Results) that both tufted and periglomerular cells increase in number, but Figure 3C only provides supporting data for periglomerular cells.

We apologize to the reviewer for the poor resolution of the PDF photos for periglomerular cells. Pictured areas were also too restricted. We, therefore, replaced the figures with high resolution photos that cover larger areas of the OB (Figure 3C and Figure 3—figure supplement 1C).

Reviewer #2:1) It is not clear exactly what the relationship is between the oxytocin experiments and the rest of this study. First, do oxytocin mutants show alterations Sema7a expression, or PlexC1 temporal distribution in the tuft structure? Why is a different behavioral assay used for the oxytocin experiments as compared to the social interaction assay used in the Plexc1 conditional mutants.Overall, there remain questions regarding the behavioral experiments such that if the oxytocin experiment was not part of this study, it would not in my view reduce interest or impact of the work.

We apologize that the relationship between the Sema7A/PlxnC1 signaling and oxytocin experiments were not clearly described in the manuscript. This was also pointed out by the other reviewers. We observed that if the pups are exposed to a particular odorant during the critical period, they demonstrate increased sensitivity and positive responses to the conditioned odor as adults. We found that Sema7A and its receptor PlxnC1 are involved in this imprinting. Odor imprinting appears to be established by enlarging the responded glomeruli during the neonatal critical period, which is accomplished by promoting post-synaptic events within the glomeruli. In the gain-of-function and loss-of-function experiments, it became clear that Sema7A/PlxnC1 signaling is essential for imprinting (increase in the sensitivity to the conditioned odorant). However, it was not clear to us whether the Sema7A/PlxnC1 signaling is also involved in imposing positive quality on the imprinted odor.

We, therefore, searched for candidate hormones expressed in the neonatal brain that may be responsible for conditioned learning and found that oxytocin KO mice fail to demonstrate positive responses to imprinted odorants. As examples, lasting interest in the imprinted odor is not seen in the habituation/dishabituation test (Figure 7A) and stress in the KO is not eased by imprinted odor memory as observed in the hyperthermia test (Figure 7B). Interestingly, however, increase in the sensitivity can still be found for the conditioned odor.

As advised by the reviewer, we confirmed that Sema7A expression is not affected in the oxytocin KO mice (Figure 7—figure supplement 1). Our results indicate that Sema7A signaling is not downstream of oxytocin. Taken together, Sema7A is responsible for imprinting, while oxytocin is separately involved in imposing quality onto imprinted memory without an effect on Sema7A function. These are now described in the text and additional data are shown in Figure 7 and Figure 7—figure supplement 1.

The reviewer also points out why the behavioral assay used for oxytocin (old Figure 6) is different from that for PlxnC1 cKO. Separately from other social experiments, the social memory test for oxytocin KO was performed in a different laboratory in collaboration with Dr. Kikusui’s group at Azabu University. In attempts to link these two sets of experiments, we added new data of the stress-reducing experiments for both the oxytocin KO and PlxnC1 cKO (Figures 6D and 7B).

2) Does loss of Sema7a in ORs lead to the same behavioral phenotypes observed in the cKO M/T PlexC1 mutants? And is there any way to show that the loss-of-function (LOF) experiment involving PlexC1 is dependent upon LOF during the critical period? For this current study, addressing the Sema7a requirement genetically in this behavioral assay seems to be an issue that should be tackled in the context of the social interaction assay, if it is to be presented here.

The reviewer asks whether we could obtain the same behavioral phenotypes in the Sema7A KO as observed in the PlxnC1 cKO. We agree that this is an important point. However, the Sema7A KO used in our study is a total KO and cannot be used for behavioral experiments as Sema7A is also expressed in various brain areas other than the olfactory system. Thus, we would need to generate the cKO of Sema7A to address the reviewer’s comment. At least for synapse formation and dendrite selection of M/T cells, effects of PlxnC1 cKO are essentially the same as that of Sema7A KO (Inoue et al., 2018). This is because both molecules are membrane-bound proteins and function together by interacting with one another.

The reviewer also suggests interesting experiments of loss-of-function and gain-of-function for PlxnC1 by changing its localization time-frame. This is an elegant experiment to link the functions of PlxnC1 at the molecular and behavioral levels. It would be nice if we could see the shift of time window for the olfactory critical period. One difficulty here is that it would not be easy to regulate not only the expression of PlxnC1, but also its localization to M/T-cell dendrites. This experiment would require new knock-in mice containing the plasmid that directs on and off of PlxnC1 localization at various time points. We would like to incorporate this experiment in our future project. In the revised manuscript, the reviewer’s point is mentioned in the text.

3) I do not have many comments on the first part of the study, however, and find it to be quite interesting. I do wonder:a) If in the conditional CNG mutants, whether or not there is any change in the temporal localization of PlexC1 with respect to the presented postnatal localization in the tuft structure-is this, too, activity dependent and/or regulated by Sema7a during the critical period.b) Is there any evidence that Sema7a OR expression normally follows a time course that directly links it to the critical period. This seems like a point that could be easily addressed.

The reviewer asks whether the PlxnC1 localization is affected by OR activity or by Sema7A expression in OSNs. We are finding that PlxnC1 localization is not affected in the Sema7A KO or in the CNG-A2 KO where Sema7A expression is severely lowered.　Although Sema7A and PlxnC1 function together by interacting with one another, PlxnC1 expression is not affected by its partner molecule. These are now stated in the text and shown in Figure 4—figure supplement 1B.

The reviewer then asks whether there is any evidence that Sema7A and OR expression follow a time course linking them to the critical period. Each OR species has own onset time for its expression during development, which is restricted to the time frame of the critical period. Also, Sema7A or oxytocin expression is not restricted to the first week after birth. Among the three key molecules (Sema7A, PlxnC1, and oxytocin) involved in imprinting, only PlxnC1’s localization to M/T-cell dendrites is directly correlated to the time frame of the olfactory critical period. This is now mentioned in the text.

Reviewer #3:1) The conclusion of the paper appears to be based on the following logical connection: activity driven changes in synaptic function in the olfactory bulb is restricted to the first week after birth because activities drive the expression of Sem7A expression, which interacts with PlxnC1, expressed only during the critical period. This is not explicitly stated, nor extensively discussed. Although the model is very attractive, the experimental evidence is correlational. The conclusion that Sema7A/PlxnC1 support the olfactory critical period is largely based on the restricted dendritic expression PlxnC1 during the first postnatal week.

There are three major molecules involved in olfactory imprinting in neonates: They are (1) Sema7A that supports odor-induced activity dependency of the imprinting process, (2) PlxnC1 that is a receptor of Sema7A, which restricts the time frame of the critical period to the first week after birth, and (3) Oxytocin that is needed in imposing the positive quality on imprinted odor memory. As also pointed out by the other reviewers, we agree that functional connections of these molecules are not explicitly stated nor extensively discussed. In response to these comments, we added supporting information in the text.

2) Naris occlusion and odor exposure experiments are complementary to each other to provide support for activity dependent regulation of Sema7A expression, but there is a discrepancy in the results. Naris occlusion experiment show that the level of synaptic proteins is normal if occlusion is reopened before P7. However, odor exposure using vanillin between P2 and P4 altered odor habituation and stress response. The authors have not explained the discrepancy in the experimental design and results with regard of the difference in timing.

We apologize that this point was not carefully explained in the text. In neonates, they need to be exposed to environmental odors to make the olfactory system functional. If such stimuli are blocked by naris occlusion throughout the critical period, odor perception is affected as adults. By changing the re-open timing of occluded naris started at P0, we determined the critical period to be from P0 to P7.

We also performed a gain**-**of-function experiment by exposing pups to a particular odorant. Unlike the naris occlusion experiment, this experiment does not examine all glomeruli as a whole, but only a particular OR. Since the onset of OR genes varies among different OR species, timing for gain-of-function (glomerular enlargement/imprinting) is different from glomerulus to glomerulus. Furthermore, synapse formation occurs using not only odor-evoked but also intrinsic OR activity. Since the early-onset OR with low intrinsic activity is convenient for our gain-of-function experiment, we chose MOR29A whose ligand is known as VNL. We found that exposure to VNL only for a few days during the critical period enhanced synapse formation in the responding glomeruli and increased the sensitivity to VNL. In the case of VNL-responsive MOR29A, P2~4 was most efficient for imprinting, but P5~7 was much less effective (Figure 3B and D).

The reviewer points out the discrepancy between the loss-of-function and gain-of-function experiments for the timing of critical period. It should be noted that in the loss-of-function experiment (naris occlusion), the critical period covers the maximum time frame for all OR species, while in the gain-of-function experiment, (odor exposure) critical periods are narrower and different among OR species depending upon their onset. These are now explained in the text.

3) The CNGA2 KO experiments lend support for activity-dependent expression of Sema7A, and the Sema7A rescue experiment provides further support that Sema7A could induced synaptic protein expression. However, different glomeruli exhibit different levels of Sema7A expression. Is this difference intrinsic to different types of OSNs, or is it caused by differences in neural activity?

Our experiments of KO, naris occlusion, and odor exposure all support the involvement of OR-derived neuronal activity in synapse formation mediated by Sema7A. As pointed out, it was not made clear in our previous manuscript whether the OR-derived activity is intrinsic or odor-evoked. We assume that both types of activities are involved in synapse formation within glomeruli. Since unique levels of Sema7A are detected in each glomerular species in unstimulated mice (please see glomerular ranking for Sema7A expression in Figure 4B), intrinsic OR activity appears to be responsible for forming basic circuitry.

However, synapses are further strengthened and glomerular structures enlarged by odor-evoked OR activity in neonates. This plastic process in neonates induced by environmental odors is responsible for the establishment of imprinted odor memory during the critical period. Thus, different levels of Sema7A, as shown in the glomerular rank in the unstimulated mice, are due to the differences in intrinsic OR activity, while differences in Sema7A expression in the odor-stimulated glomeruli are due to odor-evoked neural activity. These are now described is the Results and discussed more in the Discussion.

4) The conditional PlxnC1 KO mice need more rigorous testing. If activity-dependent expression of Sema7A and Sema7A/PlxnC1 interaction are important for synapse formation in the olfactory glomeruli, odor detection and discrimination are expected to be affected. No experiments have been conducted to address this prediction. It is also not known whether the KO abolishes the phenotypes observed in Figure 2, which are associated with vanillin exposure during the critical period. Moreover, it is not clearly stated how the three-chamber assay is related olfactory imprinting. A likely explanation of the observed impairment of odor preference and social behavior is that the overall olfactory sense has been compromised.

PlxnC1 cKO failed to demonstrate stress-reducing effect to VNL as adults in the hyperthermia test (Figure 6D). In the three-chamber test, WT mice demonstrate curiosity responses to foreign mouse scents, but the PlxnC1 cKO avoid them (Figure 6C). It is possible that the mice innately avoid foreign mouse scents that may be stressful to them, but odor-induced imprinting helps them feel comfortable to the surrounding mice for smooth social interactions. We assume that this is due to the lack of odor imprinting mediated by odor-evoked OR-activity.

As pointed out, in the PlxnC1 cKO where Sema7A signaling is blocked, odor detection and discrimination are lowered (Figure 6A). However, it is notable that these mice still demonstrate attractive responses to food smells and aversive behavior toward foreign mouse scents (Figure 6B). It appears that even in the absence of Sema7A signaling in the cKO, basic olfactory circuits, particularly to induce innate responses, can be established based on a genetic program, e. g., formation of gap junctions and connections to interneurons (Gire et al., 2012). In the CNG-A2 KO, synapse formation and dendrite selection within glomeruli are initially impaired, but later recover presumably using intrinsic OSN activity.

We, therefore, do not think that the observed impairment in odor preference and social behavior is due to the overall compromise of olfactory sensing as the reviewer points out. As we agree that this is an important point, we added some discussion in the new text.

5) The oxytocin KO experiment is interesting. However, the behavioral phenotype is difficult to interpret. WT mice exhibit a decline in investigation of an ovariectomized female whereas the oxy-/- mice show persistent investigation. The authors interpret the decline as an indication of social memory, but the lack of decline/persistent investigation in the oxy-/- mice could be interpreted as prosocial interactions. This interpretation would be consistent with the decrease of investigation in WT mice after P0-P6 oxytocin injection. Importantly, the authors have not addressed how this experiment is related to activity-dependent modulation of glomerular connectivity.

In the social-memory test shown in Figure 8, WT mice exhibit a decline in investigation of female mice while the oxytocin KO demonstrates persistent investigation. We assume that this is due to the lack of positive imprinting to become friendly to fellow mice, which is mediated by oxytocin in neonates. Oxytocin KO is not accustomed to the unfamiliar mouse scents in adults. This is supported by our observation that in the oxytocin KO, the sensitivity to the imprinted odor is increased but the positive quality is not imposed on the conditioned odor (Figure 7). Thus, Sema7A/PlxnC1 signaling is needed for imprinting (increase in the sensitivity to the imprinted odor), but oxytocin is separately needed to add the positive quality on the imprinted memory of mouse scents to establish social memory. These are now described in the text.

6) The authors present data with quantification, but primary data (images) are not presented. This makes it difficult to assess the accuracy of the quantification. Specifically, primary images demonstrating the selective induction of Sema7A expression in the MOR29A glomeruli after vanillin exposure should be compared with no-odor control side-by-side.

The reviewer points out that primary data are needed to assess the accuracy of the quantification. We now added primary images (Figure 1—figure supplement 1, Figure 3—figure supplement 1, Figure 4—figure supplement 1, and Figure 5—figure supplement 2), specifically for the selective induction of Sema7A in the VNL-exposed glomeruli (Figure 3—figure supplement 1A).